# Gravitational thermodynamics of causal diamonds in (A)dS

Ted Jacobson[1]⋆ and Manus Visser[2]†

**1** Maryland Center for Fundamental Physics,
University of Maryland, College Park, MD 20742, USA
**2** Institute for Theoretical Physics, University of Amsterdam,
1090 GL Amsterdam, The Netherlands

⋆ jacobson@umd.edu, † m.r.visser@uva.nl

## Abstract

The static patch of de Sitter spacetime and the Rindler wedge of Minkowski spacetime are causal diamonds admitting a true Killing field, and they behave as thermodynamic equilibrium states under gravitational perturbations. We explore the extension of this gravitational thermodynamics to all causal diamonds in maximally symmetric spacetimes. Although such diamonds generally admit only a conformal Killing vector, that seems in all respects to be sufficient. We establish a Smarr formula for such diamonds and a "first law" for variations to nearby solutions. The latter relates the variations of the bounding area, spatial volume of the maximal slice, cosmological constant, and matter Hamiltonian. The total Hamiltonian is the generator of evolution along the conformal Killing vector that preserves the diamond. To interpret the first law as a thermodynamic relation, it appears necessary to attribute a negative temperature to the diamond, as has been previously suggested for the special case of the static patch of de Sitter spacetime. With quantum corrections included, for small diamonds we recover the "entanglement equilibrium" result that the generalized entropy is stationary at the maximally symmetric vacuum at fixed volume, and we reformulate this as the stationarity of free conformal energy with the volume *not* fixed.


# 1 Introduction

Horizon thermodynamics was first discovered in the context of black holes [1–3], but the principles are far more universal than that. The case of cosmological horizons was quickly understood [4,5] and, rather less quickly insofar as entropy is concerned, that of acceleration horizons as well [6–11]. Most recently, in the setting of AdS/CFT duality, the gravitational thermodynamics of "entanglement wedges" has been discovered [12, 13]. An entanglement wedge is the domain of dependence of a partial Cauchy surface of the bulk, whose boundary consists of a subregion $\mathcal{R}$ of a conformal boundary Cauchy slice together with the minimal area bulk surface that meets the conformal boundary at $\partial \mathcal{R}$. The area $A$ of the minimal surface corresponds to the CFT entanglement entropy of the subregion $\mathcal{R}$ to leading order in Newton's constant, via the Ryu-Takayanagi formula, which is nothing but the Bekenstein-Hawking

entropy $A/4\hbar G$ [14,15]. In particular, a link has been established between fundamental properties of CFT entanglement entropy and the bulk Einstein equation, drawing a connection between the behavior of quantum information and gravitational dynamics in this holographic setting [16–19].

In the examples just described, the thermodynamic system extends to a boundary of spacetime. Quasilocal relations analogous to the laws of thermodynamics have been found for various sorts of 'apparent' horizons, but of course these find application only when such horizons are present [20]. If the lesson from all we have learned is that something fundamentally statistical underlies gravitational dynamics, then that something should be at play *everywhere* in spacetime. This viewpoint was the motivation for introducing the notion of "local causal horizon," with which it was possible to derive the Einstein equation from the Clausius relation applied to the area-entropy changes of all such horizons in spacetime [8]. Essential to that argument was the fact that the near vicinity of any spacetime point looks like part of a slightly deformed Minkowski spacetime, so that one can identify an approximate local boost Killing field with respect to which the relevant notion of energy can be defined. While that derivation was reasonably plausible on physical grounds, the "system" under consideration was not sharply defined. It was taken to be a very small, near horizon subsystem, but no precise definition was given, and possible effects of entanglement with neighboring regions were not addressed.

To localize and isolate it, the thermodynamic system was defined in [21] as the spacetime inside a causal diamond, i.e. the domain of dependence of a spatial ball.[1] For such a system, rather than describing a time dependent physical process, one can just consider variations, comparing the equilibrium state to nearby states. A link was established between the semiclassical Einstein equation, and the stationarity of the total entanglement entropy of the diamond with respect to variations of the geometry and quantum fields away from the vacuum state at fixed volume, taking the diamond much smaller than the local curvature length scale of the background spacetime.

A *classical* ingredient of this argument is a "first law of causal diamonds," which relates variations, away from flat spacetime, of the area, volume, and matter energy-momentum tensor inside. This relation is similar to the first law of black hole mechanics, and was derived in an Appendix of [21] using the same methods as employed by Wald for black holes [26]. A diamond in Minkowski spacetime plays the role of equilibrium state in this first law. Unlike a black hole, however, a causal diamond is only *conformally stationary*. That is, it does not admit a timelike Killing vector, but it admits a timelike conformal Killing vector, for which the null boundary is a conformal Killing horizon [27,28], with a well-defined surface gravity.[2] Remarkably, this is sufficient to derive the first law, but an additional contribution arises, namely, the gravitational Hamiltonian, which is proportional to the spatial volume of the ball.

In the present paper we generalize the first law of causal diamonds to (Anti-)de Sitter spacetime – i.e. it applies to any maximally symmetric space – and we include variations of the cosmological constant and matter stress-energy, both using a fluid description as done for black holes by Iyer [30]. Treated this way, the matter can contribute to a first order variation away from a maximally symmetric spacetime.[3] Motivation for varying the cosmological constant is discussed in Sec. 3.3.3. The first law we obtain for Einstein gravity minimally coupled to fluid

---

[1] The thermal behavior of conformal quantum fields – without gravity – inside a causal diamond in flat space was studied earlier, for example in Refs. [12, 22–25].

[2] It was shown in [29] that the surface gravity $\kappa$ of a conformal Killing horizon has a conformal invariant definition (see Appendix C for an explanation, and for a proof of the zeroth law). This fact can be viewed as a hint that conformal Killing horizons have thermodynamic properties, since $\kappa$ is identical to the surface gravity of a conformally related true Killing horizon (for which $\hbar\kappa/2\pi$ is the Hawking temperature).

[3] Matter described by a typical field theory action could not contribute, since the matter fields would vanish in the maximally symmetric background spacetime, and the action is at least quadratic in the fields.

matter takes the form

$$\delta H_\zeta = -\frac{\kappa}{8\pi G}\delta A, \tag{1.1}$$

where $\zeta$ is the conformal Killing vector of the unperturbed, maximally symmetric diamond (computed explicitly in Appendix A), $H_\zeta$ is the total Hamiltonian for gravity and matter along the flow of $\zeta$, and $\kappa$ is the surface gravity (defined as positive) with respect to $\zeta$. Much as for black holes, if we multiply and divide by Planck's constant $\hbar$, the quantity on the right hand side of (1.1) takes the form $-T_H\delta S_{BH}$, where $T_H = \hbar\kappa/2\pi$ is the Hawking temperature and $S_{BH} = A/4\hbar G$ is the Bekenstein-Hawking entropy. Unlike for black holes, however, there is a *minus sign* in front, indicating that the temperature is *negative*. This minus sign is familiar from the limiting case in which the diamond consists of the entire static patch of de Sitter (dS) spacetime, bounded by the de Sitter horizon [5]. It was argued in the dS case that the temperature of the static patch is therefore negative [31], and further arguments in favor of this interpretation were given recently in [32].

The left hand side of (1.1) consists of minimally coupled matter, cosmological constant, and gravitational terms, i.e. $\delta H_\zeta = \delta H_\zeta^{\tilde{m}} + \delta H_\zeta^{\Lambda} + \delta H_\zeta^{g}$. The contribution of the fluid matter with arbitrary equation of state is

$$\delta H_\zeta^{\tilde{m}} = \int_\Sigma \delta(T_a{}^b)^{\tilde{m}}\zeta^a u_b\, dV, \tag{1.2}$$

where $(T_a{}^b)^{\tilde{m}} = g^{bc}T_{ac}^{\tilde{m}}$ is the Hilbert stress-energy tensor with one index raised by the inverse metric and $\Sigma$ is the maximal slice of the unperturbed diamond, with future pointing unit normal vector $u^a$ and proper volume element $dV$. The cosmological constant can be thought of as a perfect fluid with stress-energy tensor $T_{ab}^{\Lambda} = -\Lambda/(8\pi G)g_{ab}$. Since it is maximally symmetric, it may be nonzero in the background. Its Hamiltonian variation is given by

$$\delta H_\zeta^{\Lambda} = \frac{V_\zeta}{8\pi G}\delta\Lambda, \qquad \text{with} \qquad V_\zeta := \int_\Sigma |\zeta|dV, \tag{1.3}$$

where $|\zeta| := \sqrt{-\zeta\cdot\zeta}$ is the norm of the conformal Killing vector. The quantity $V_\zeta$ is called the "thermodynamic volume" [33–35], and is well known in the context of the first law for black holes, when extended to include variations of $\Lambda$. Finally, the gravitational term takes the form

$$\delta H_\zeta^{g} = -\frac{\kappa k}{8\pi G}\delta V, \tag{1.4}$$

where $k$ is the trace of the (outward) extrinsic curvature of $\partial\Sigma$ as embedded in $\Sigma$, and $V$ is the proper volume of the ball-shaped spacelike region $\Sigma$. This term arises because $\zeta$ fails to be a true Killing vector. In the special case of the static patch of dS space, $\zeta$ is a true Killing vector, and indeed (1.4) vanishes, since we have $k = 0$. That this gravitational Hamiltonian variation is proportional to the maximal volume variation suggests a connection with the *York time Hamiltonian* [36], which generates evolution along a constant mean curvature foliation and is proportional to the proper volume of slices of this foliation (see Sec. 3.3.4). In fact, we show in Appendix B that slices of the constant mean curvature foliation of a maximally symmetric diamond coincide with slices of constant conformal Killing time.

Moreover, we extend the first law of causal diamonds to the semiclassical regime, i.e. by considering quantum matter fields on a fixed classical background. For any type of quantum matter in a small diamond we show that the semiclassical first law can be written in terms of

Bekenstein's generalized entropy[4]

$$-\frac{\kappa k}{8\pi G}\delta V + \frac{V_\zeta}{8\pi G}\delta\Lambda = T\delta S_{\text{gen}}, \qquad (1.5)$$

where the temperature is minus the Hawking temperature, i.e. $T = -\hbar\kappa/2\pi$, and the generalized entropy is defined as the sum of the Bekenstein-Hawking entropy and the matter entanglement entropy, i.e. $S_{\text{gen}} := S_{\text{BH}} + S_{\tilde{\text{m}}}$. At fixed volume $V$ and cosmological constant $\Lambda$ this implies that the generalized entropy is stationary in a maximally symmetric vacuum. This coincides with the *entanglement equilibrium* condition of [21], which was shown in that paper to be equivalent to the semiclassical Einstein equation. We also argue in Sec. 4.4 that the entanglement equilibrium condition is equivalent to the stationarity of a free energy at fixed cosmological constant, but *without* fixing the volume. We thus find that the semiclassical Einstein equation is also equivalent to the stationarity of free energy.

The present paper complements other investigations of causal diamonds in flat space [39–41]. In particular, the first law of causal diamonds was generalized to higher derivative gravity in [42], in which case the Bekenstein-Hawking entropy should be replaced by the Wald entropy and the proper volume by the "generalized volume" $W$. That first law is also valid in any maximally symmetric space, but for (A)dS space the derivation was based on certain identities which we prove below, in particular (2.8) and (2.13). Further, in [43] the second order area variation of a small geodesic ball was computed in the absence of matter and compared with the gravitational energy. This is a step towards an extension of the first law of causal diamonds to second order variations. Ref. [44] derives a Clausius relation for the reversible part of the entropy change between time slices of causal diamonds in flat space, and compares that to the first law for causal diamonds in higher curvature gravity. Finally, in [45] the four laws of thermodynamics were established for the causal complement of a diamond in flat space, in particular a physical process version of the first law was derived. This differs from our first law, in the sense that the latter is an equilibrium state version.

This paper is organized as follows. In Sec. 2 we describe our setup of causal diamonds in maximally symmetric spaces in more detail. The Smarr formula and first law for causal diamonds are derived in Sec. 3 using Wald's Noether charge formalism. In Sec. 4 we give a thermodynamic interpretation to the first law, and we derive the entanglement equilibrium condition from the semiclassical first law. In Sec. 3.3 we comment on various aspects of the first law, and in Sec. 5 we describe a number of limiting cases of the first law for maximally symmetric causal diamonds: de Sitter static patch, flat space, Rindler space, AdS-Rindler space and the Wheeler-DeWitt patch of AdS. We end with a summary of results and a discussion of possible future research directions in Sec. 6. The appendices are devoted to establishing several properties of conformal Killing fields in (A)dS and in flat space, and of bifurcate conformal Killing horizons in general.

## 2 Causal diamonds in maximally symmetric spaces

In this section we discuss causal diamonds and their conformal Killing vectors in maximally symmetric spaces. It suffices to write the equations for the case of positive curvature, i.e. de Sitter space, since the negative curvature (Anti-de Sitter) and flat cases can be obtained from these by sending the curvature length scale $L$ to $iL$, or to infinity, respectively. The line element

---

[4]For conformal matter this expression for the first law actually holds for any sized diamond, whereas for non-conformal matter the derivation depends on an assumption (4.13) about the modular Hamiltonian variation for small diamonds which was conjectured in [21] and tested in [37, 38].

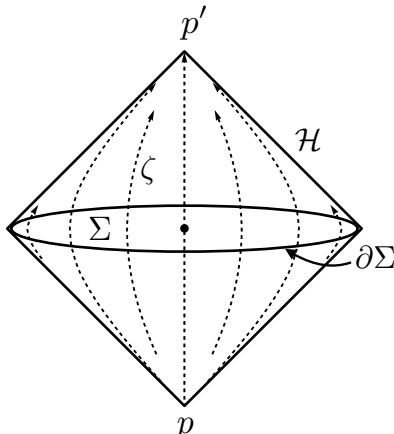

Figure 1: A causal diamond in a maximally symmetric spacetime for a ball-shaped spacelike region $\Sigma$. The past and future vertices of the diamond are denoted by $p$ and $p'$, respectively, and $\mathcal{H}$ is the null boundary. The dashed arrows are the flow lines of the conformal Killing vector $\zeta$, whose flow sends the boundary of the diamond into itself, and vanishes at $\partial\Sigma$, $p$ and $p'$.

for a static patch of de Sitter space in $d$ spacetime dimensions is

$$ds^2 = -[1-(r/L)^2]dt^2 + [1-(r/L)^2]^{-1}dr^2 + r^2 d\Omega_{d-2}^2, \tag{2.1}$$

where we use units with $c = 1$. In terms of retarded and advanced time coordinates,

$$u = t - r_* \qquad \text{and} \qquad v = t + r_*, \tag{2.2}$$

with $r_*$ the "tortoise coordinate" defined by

$$dr_* = \frac{dr}{1-(r/L)^2}, \qquad r = L\tanh(r_*/L), \tag{2.3}$$

the line element (2.1) takes the form

$$\begin{aligned}
ds^2 &= -[1-(r/L)^2]\,du\,dv + r^2 d\Omega_{d-2}^2 \\
&= \text{sech}^2(r_*/L)\big[-du\,dv + L^2\sinh^2(r_*/L)\,d\Omega_{d-2}^2\big].
\end{aligned} \tag{2.4}$$

Note that $r_* = r + O(r^3)$, so in particular $r = 0 = r_*$ at the origin. For dS the cosmological horizon, $r = L$, corresponds to $r_* = \infty$. For AdS, we have $r = L\tan(r_*/L)$, so $r = \infty$ corresponds to $r_* = L\pi/2$. In the flat space limit, $L \to \infty$, the tortoise and radial coordinates coincide.

A spherical causal diamond in a maximally symmetric space can be defined as the domain of dependence of a spherical spacelike region $\Sigma$ with vanishing extrinsic curvature. Equivalently, it can be described as the intersection of the future of some point $p$, with the past of another point $p'$ (see Fig. 1). All such diamonds are equivalent, once the geodesic proper time between the *vertices* $p$ to $p'$ has been fixed. The intersection of the future light cone of $p$ and the past light cone of $p'$ is called the *edge* of the diamond. The edge is the boundary $\partial\Sigma$ of a $(d-1)$-dimensional ball-shaped region $\Sigma$. The symmetries of such diamonds are rotations about the $pp'$ line, reflection across $\Sigma$, and a conformal isometry to be discussed shortly.

Fixing a causal diamond, we place the origin of the above coordinate system at the center, and choose the $t$ coordinate so that $\Sigma$ lies in the $t = 0$ surface. The geodesic joining the vertices is then the line $r = 0$, and the diamond is the intersection of the two regions $u > -R_*$ and $v < R_*$, for some $R_*$. The vertices are located at the points $p = \{u = v = -R_*\}$ and

$p' = \{u = v = R_*\}$, and the edge of the diamond is the $(d-2)$-sphere $\partial\Sigma = \{v = -u = R_*\}$, with coordinate radius $r_* = R_*$ and area radius $r = R = L\tanh(R_*/L)$.

The unique conformal isometry that preserves the causal diamond is generated by the conformal Killing vector

$$\zeta = \frac{L}{\sinh(R_*/L)}\Big[\big(\cosh(R_*/L) - \cosh(u/L)\big)\partial_u + \big(\cosh(R_*/L) - \cosh(v/L)\big)\partial_v\Big]. \quad (2.5)$$

A derivation of this fact is given in Appendix A. The flow generated by $\zeta$ sends the boundary of the diamond into itself, and leaves fixed the vertices and the edge. In the interior $\zeta$ is the sum of two future null vectors, so it is timelike and future directed. It is the null tangent to the past and future null boundaries, $\mathcal{H} = \{v = R_*, u = -R_*\}$, so those boundaries are conformal Killing horizons. The intersection of these null boundaries, i.e. the edge $\partial\Sigma$, is therefore referred to as the *bifurcation surface* of $\mathcal{H}$.[5]

In terms of the $t$ and $r$ coordinates introduced above, the conformal Killing vector reads

$$\zeta = \frac{L^2}{R}\left[\left(1 - \frac{\sqrt{1-(R/L)^2}}{\sqrt{1-(r/L)^2}}\cosh(t/L)\right)\partial_t - \frac{r}{L}\sqrt{(1-(R/L)^2)(1-(r/L)^2)}\sinh(t/L)\,\partial_r\right].$$
$$(2.6)$$

Note that if the boundary of the diamond coincides with the cosmological horizon, i.e. if $R = L$, then $\zeta = L\partial_t$. That is, the conformal Killing symmetry becomes the usual time translation of the entire static patch of dS, which is a causal diamond with infinite time duration but finite spatial width. In Appendix D we use the two-time embedding formalism of dS and AdS to derive an expression for $\zeta$ in terms of the generators of the conformal group.[6] We show that $\zeta$ can be written as a linear combination of a time translation of the surrounding static patch and a conformal transformation (which is a special conformal transformation in the case of flat space) – see equation (D.8).

The surface gravity $\kappa$ of any conformal Killing vector with a bifurcation surface $\mathcal{B}$ can be defined exactly as for a true Killing vector: If we contract the conformal Killing equation, $\nabla_{(a}\zeta_{b)} = \alpha g_{ab}$ with $m^a m^b$, for $m^a$ any tangent vector to $\mathcal{B}$, the left hand side vanishes, since $\zeta^a = 0$ on $\mathcal{B}$. Thus we learn that (as for the particular example of $\zeta^a$ under study) $\alpha = 0$ on $\mathcal{B}$. It follows that $\partial_a\zeta^b$ is a generator of Lorentz transformations in the two-dimensional normal plane at each point of $\mathcal{B}$. Like for true Killing vectors, we may therefore define the surface gravity $\kappa$ at $\mathcal{B}$ (or rather its absolute value) by $2\kappa^2 = (\partial_a\zeta^b)(\partial_b\zeta^a)$. Other definitions for $\kappa$, which are equivalent for a true Killing vector, are not equivalent for a conformal Killing vector. The definition that is invariant under conformal rescalings of the metric [29], $\nabla_a\zeta^2 = -2\kappa\zeta_a$, is also constant along the generators of the conformal Killing horizon, and coincides with the definition just given at $\mathcal{B}$. We establish these general properties in Appendix C, along with the zeroth law, i.e. the fact that $\kappa$ is constant on $\mathcal{H}$. The normalization of $\zeta^a$ in (2.5) or (2.6) has been chosen so that $\kappa = 1$ at the future horizon ($v = R_*$) and $\kappa = -1$ at the past horizon ($u = -R_*$). In the rest of the paper we take $\kappa$ to be positive and keep it explicit, to indicate where it appears if a different normalization is chosen.[7]

---

[5]The conformal Killing vector also acts outside the diamond, and remains null on the continuation of the null boundaries of the diamond. In total there are four conformal Killing horizons $\{u = \pm R_*, v = \pm R_*\}$, which divide the maximally symmetric spacetime up into six regions (see [45] for an analysis of the flow of the conformal Killing field in Minkowski spacetime).

[6]Note that maximally symmetric spaces are conformally flat, so they admit the $O(2,d)$ group of conformal isometries.

[7]If the conformal Killing vector were normalized such that $\zeta^2 = -1$ at $t = r = 0$, then the surface gravity would be $\kappa = (R/L^2)(1-\sqrt{1-(R/L)^2})^{-1}$. This is the choice usually made in the case of the static patch of de Sitter space, where $R = L$ and $\kappa_{\mathrm{dS}} = 1/L$. In the flat space limit ($L \to \infty$) this surface gravity becomes $\kappa_{\mathrm{flat}} = 2/R$, and for an infinite diamond in AdS ($R/L \to \infty$, the "Wheeler-DeWitt patch") it reduces to $\kappa_{\mathrm{WdW}} = 1/L$, where $L$ is the AdS radius.

The Lie derivative of the metric with respect to a conformal Killing vector in general has the form

$$\mathcal{L}_\zeta g_{ab} = \nabla_a \zeta_b + \nabla_b \zeta_a = 2\alpha g_{ab}, \qquad \alpha = (\nabla_c \zeta^c)/d. \tag{2.7}$$

In addition, $\zeta$ has some special properties that will be important for us. Under the reflection symmetry of the diamond, $\zeta \to -\zeta$, so $\alpha \to -\alpha$. It follows that $\alpha$ vanishes on $\Sigma$, so $\zeta$ acts "instantaneously" as a true Killing vector on $\Sigma$:

$$\mathcal{L}_\zeta g_{ab}\big|_\Sigma = 0. \tag{2.8}$$

It also follows that $\nabla_a \alpha$ is normal to $\Sigma$, so we have

$$\nabla_c \big(\mathcal{L}_\zeta g_{ab}\big)\big|_\Sigma = -2\dot\alpha\, u_c g_{ab}, \qquad \dot\alpha = u^c \nabla_c \alpha, \tag{2.9}$$

where $u^c$ is the future pointing unit normal to $\Sigma$, given by $u^c \partial_c = \big(1 - (r/L)^2\big)^{-1/2} \partial_t$. Computation of $\alpha$ using the expression (2.6) easily yields

$$\alpha = -\frac{L}{R}\sqrt{(1-(R/L)^2)(1-(r/L)^2)}\sinh(t/L), \tag{2.10}$$

$$\dot\alpha\big|_{t=0} = -\frac{1}{R}\sqrt{1-(R/L)^2} = \frac{-1}{L\sinh(R_*/L)}. \tag{2.11}$$

Notice that $\dot\alpha$ is *constant* on $\Sigma$. This property will be crucial for the existence of a geometric form of the first law of causal diamonds. Further, since

$$k := \frac{d-2}{R}\sqrt{1-(R/L)^2} \tag{2.12}$$

is the trace of the extrinsic curvature of the surface $\partial\Sigma$ as embedded in $\Sigma$, we may write

$$\dot\alpha\big|_\Sigma = -\frac{\kappa k}{d-2}, \tag{2.13}$$

allowing for a normalization of $\zeta$ with surface gravity $\kappa$ rather than unity.

While we established (2.13) by explicit computation, it can also be derived by examining derivatives along the horizon at $\partial\Sigma$, and using the properties that (i) the trace of the extrinsic curvature of $\Sigma$ vanishes, and (ii) $\dot\alpha$ is constant on $\Sigma$. However, we have not found an underlying geometric reason for the constancy of $\dot\alpha$ on $\Sigma$. It probably requires maximal symmetry of the spacetime, since we checked that $\dot\alpha$ is not constant for the case of a causal diamond in time cross hyperbolic space, $\mathbb{R} \times \mathbb{H}^{d-1}$, which also admits a diamond preserving conformal Killing vector. In Appendix B the constancy of $\dot\alpha$ on $\Sigma$ is established in a different way.

## 3 Mechanics of causal diamonds in (A)dS

In this section we first derive a Smarr formula for causal diamonds in (A)dS, by equating the Noether charge to the integral of the Noether current. This method of obtaining the Smarr formula is quite general, and it illustrates the origin of the "thermodynamic volume" term. As an aside, we also show how a finite Smarr formula for AdS black holes can be obtained by subtracting the (divergent) empty AdS Smarr formula from the black hole one. We then move on to our main objective, which is to derive the first law of causal diamonds in (A)dS. First we employ the usual dimensional scaling argument to deduce from the Smarr formula a first law for variations between maximally symmetric diamonds. Next we employ the Noether current method, used by Wald for the case of black holes [26, 46], as was done for variations of Minkowski space diamonds in Appendix D of [21]. Here we extend that derivation to (Anti-)de Sitter space, and include fluid matter (allowing in particular for a variable cosmological constant) as in [30], obtaining a first law that applies to arbitrary variations to nearby solutions.

### 3.1 Smarr formula for causal diamonds

We start with deriving a Smarr-like formula for causal diamonds in (A)dS space. We will obtain this relation using a slightly unusual but very general method: equating the Noether current associated with diffeomorphism symmetry to the exterior derivative of the Noether charge and integrating over the ball $\Sigma$. Liberati and Pacilio [47] used the same Noether method to derive a Smarr formula for Lovelock black holes. Throughout this section we will employ Wald's Noether charge formalism [26]. See e.g. [46,48] for further details about this formalism.

To every Lagrangian $d$-form depending on the dynamical fields $\phi$ there is an associated *symplectic potential* $(d-1)$-form $\theta$, defined through

$$\delta L = E\,\delta\phi + d\theta(\phi,\delta\phi), \tag{3.1}$$

where $E$ is the equation of motion $d$-form, and tensor indices are suppressed. For a variation $\delta\phi = \mathcal{L}_\chi\phi$ induced by the flow of a vector field $\chi$, there is an associated *Noether current* $(d-1)$-form,

$$j_\chi := \theta(\phi, \mathcal{L}_\chi\phi) - \chi \cdot L. \tag{3.2}$$

When $L$ is diffeomorphism covariant, the variation $\delta_\chi L$ produced by $\delta\phi = \mathcal{L}_\chi\phi$ is equal to $\mathcal{L}_\chi L$, which implies that the Noether current is closed for all $\chi$ when the equation of motion $E = 0$ holds, and so is an exact form,

$$j_\chi = dQ_\chi. \tag{3.3}$$

The $(d-2)$-form $Q_\chi$ is constructed from the dynamical fields together with $\chi$ and its first derivative, and is called the *Noether charge* form.

The Smarr formula comes from the integral version of the identity (3.3),

$$\oint_{\partial\mathcal{R}} Q_\chi = \int_{\mathcal{R}} j_\chi, \tag{3.4}$$

where $\mathcal{R}$ is a $(d-1)$-dimensional submanifold with boundary $\partial\mathcal{R}$. For a black hole with bifurcate Killing horizon, $\chi$ can be taken as the horizon generating Killing vector, and $\mathcal{R}$ can be taken as a hypersurface extending from the bifurcation surface to spatial infinity. Since $\mathcal{L}_\chi\phi = 0$ when $\chi$ is a Killing symmetry of all the dynamical fields, the first term of (3.2) vanishes. For vacuum Einstein gravity, without a cosmological constant, the second term of (3.2) vanishes on shell, so (3.4) reduces to the statement that the Noether charge of the horizon is equal to that of the sphere at spatial infinity, both orientations being taken outward (toward larger radius). This yields the Smarr formula [49],

$$\frac{d-3}{d-2}M - \Omega_{\mathcal{H}}J = \frac{\kappa}{8\pi G}A, \tag{3.5}$$

where $M$ is the mass, $J$ is the angular momentum, and $\Omega_{\mathcal{H}}$ is the angular velocity of the horizon, and $A$ is the horizon area.

For a maximally symmetric causal diamond in Einstein gravity, with or without a cosmological constant, we can instead choose the region $\mathcal{R}$ to be the ball $\Sigma$, and choose the vector field $\chi$ to be the conformal Killing vector $\zeta$ of the diamond. The left hand side of (3.4) is then just a single integral,[8]

$$\oint_{\partial\Sigma} Q_\zeta = -\frac{\kappa}{8\pi G}A. \tag{3.6}$$

---

[8]The orientation is chosen to be outward, toward larger radius, according to Stokes' theorem. The minus sign is unfamiliar, because in the black hole case the orientation of the Noether charge integral on the horizon is typically chosen, as in [26], so as to be towards spatial infinity.

On the other hand, the contribution from the integral of the Noether current on the right hand side of (3.4) no longer vanishes: since $\zeta$ is not a Killing vector, the symplectic potential term in the Noether current (3.2) is nonzero and, if the cosmological constant is nonvanishing, the Lagrangian no longer vanishes on shell so the second term in the Noether current is also nonzero. To evaluate the contribution from the first term in the Noether current, we note that the symplectic potential for Einstein gravity is given by [46, 50]

$$\theta(g, \delta g) = \frac{1}{16\pi G} \epsilon_a (g^{ab} g^{ce} - g^{ae} g^{bc}) \nabla_e \delta g_{bc},\tag{3.7}$$

where $\epsilon_a = \epsilon_{a a_2 \cdots a_d}$ is the volume form with the first index displayed and the remaining $(d-1)$ indices suppressed. Setting $\delta g_{bc} = \mathcal{L}_\zeta g_{bc} = 2\alpha g_{bc}$, evaluating on $\Sigma$, and using (2.9), we obtain

$$\theta(g, \mathcal{L}_\zeta g)|_\Sigma = \frac{(d-1)\dot{\alpha}}{8\pi G} u \cdot \epsilon,\tag{3.8}$$

and together with (2.13) this yields

$$\int_\Sigma \theta(g, \mathcal{L}_\zeta g) = -\frac{d-1}{d-2}\frac{\kappa k}{8\pi G} V,\tag{3.9}$$

where $V = \int_\Sigma u \cdot \epsilon$ is the proper volume of the ball. To evaluate the contribution from the second term in the Noether current, we note that the off-shell Lagrangian is

$$L = \frac{R - 2\Lambda}{16\pi G} \epsilon.\tag{3.10}$$

On shell we have $R - 2\Lambda = 4\Lambda/(d-2)$, so the on-shell Lagrangian is

$$L^{\text{on-shell}} = \frac{\Lambda}{(d-2)4\pi G} \epsilon,\tag{3.11}$$

and the second term in the integral of the Noether current is thus

$$-\int_\Sigma \zeta \cdot L = -\frac{\Lambda}{(d-2)4\pi G} V_\zeta,\tag{3.12}$$

where

$$V_\zeta := \int_\Sigma \zeta \cdot \epsilon.\tag{3.13}$$

Since $\zeta$ is orthogonal to $\Sigma$, $V_\zeta$ is just the proper volume of $\Sigma$ weighted locally by the norm of the conformal Killing vector, given in (1.3). For the case of a black hole in asymptotically Anti-de Sitter spacetime, a quantity close to $V_\zeta$ was first identified in [33] as the variable thermodynamically conjugate to $\Lambda$. (See subsection 3.1.1 for a discussion of that case.) For a true Killing vector it is commonly called the *thermodynamic volume* [34, 35], and we will use that term also in the conformal Killing case. For the conformal Killing vector (2.6) in dS space $V_\zeta$ is easily found to be given by

$$V_\zeta = \frac{\kappa L^2}{R} \left( V_R^{\text{flat}} - \sqrt{1 - (R/L)^2} \, V_R \right),\tag{3.14}$$

where $V_R^{\text{flat}} = R^{d-1} \Omega_{d-2}/(d-1)$ is the volume of a sphere of radius $R$ in Euclidean space and $V_R$ is the proper volume of a ball of radius $R$ in dS space. It follows from the definition (3.13) that $V_\zeta$ is positive in both dS and AdS space, although that is not obvious from the expression in (3.14). In the flat space limit $R/L \to 0$ it becomes $V_\zeta^{\text{flat}} = \kappa R V_R^{\text{flat}}/(d+1)$.

Combining the two terms (3.9) and (3.12), the integral of the Noether current is thus given by

$$\int_\Sigma j_\zeta = -\frac{(d-1)\kappa k V + 2\Lambda V_\zeta}{(d-2)8\pi G}, \tag{3.15}$$

so (3.4) yields the Smarr formula,

$$(d-2)\kappa A = (d-1)\kappa k V + 2V_\zeta\Lambda. \tag{3.16}$$

At the cosmological horizon of de Sitter space the extrinsic curvature trace $k$ vanishes, hence the Smarr formula reduces to a relation between the horizon area and the cosmological constant. In flat space the cosmological constant is zero, so that the formula turns into the trivial connection between the area and the volume. For generic sizes of the causal diamond, and for a positive cosmological constant, equation (3.16) can be checked explicitly by using the formulas (2.12) and (3.14) for $k$ and $V_\zeta$, respectively, and the expression for the cosmological constant of dS space: $\Lambda = +(d-1)(d-2)/2L^2$.

### 3.1.1 Smarr formula for AdS black holes

As an aside from the main topic of our paper, in this subsection we discuss briefly how the Smarr formula for asymptotically AdS black holes [33] can be derived using (3.4). In that setting $\chi$ is replaced by the horizon generating Killing field $\xi$, and the domain of integration $\Sigma$ is from the black hole horizon to infinity. This does not yet yield a meaningful Smarr formula, since both $\oint_\infty Q_\xi$ and $\int_\Sigma j_\xi$ diverge. However, these divergences are the same as those that arise for pure AdS, so by subtracting the pure AdS Smarr formula from the AdS black hole Smarr formula, one obtains a finite relation:

$$\oint_\infty (Q_\xi - Q_\xi^{\text{AdS}}) - \oint_{\mathcal{H}} Q_\xi = \int_\Sigma j_\xi - \int_{\Sigma'} j_\xi^{\text{AdS}}, \tag{3.17}$$

where the Noether charge integrals are both outward oriented. The domain of integration $\Sigma$ in the black hole integral on the right extends from the horizon to infinity, while in the pure AdS integral the domain $\Sigma'$ extends across the entire spacetime.

The first integral on the left hand side of (3.17) is proportional to the (AdS background subtracted) Komar mass and angular momentum, and for Einstein gravity the horizon integral is proportional to the surface gravity times the horizon area. Using (3.11) and $\theta(g, \mathcal{L}_\xi g) = 0$ for Killing vectors, we find that the Noether current is $-\Lambda \xi \cdot \epsilon/((d-2)4\pi G)$. Thus the Smarr formula for AdS black holes is [33]

$$\frac{d-3}{d-2}M - \Omega_{\mathcal{H}}J = \frac{\kappa A}{8\pi G} - \frac{2\bar{V}_\xi\Lambda}{(d-2)8\pi G}, \tag{3.18}$$

where

$$\bar{V}_\xi := \int_\Sigma \xi \cdot \epsilon - \int_{\Sigma'} \xi^{\text{AdS}} \cdot \epsilon^{\text{AdS}} \tag{3.19}$$

is the background subtracted thermodynamic volume.[9] The relative sign between the area term and cosmological constant term is the same as in the Smarr formula for causal diamonds (3.16). For AdS-Schwarzschild, however, the quantity $\bar{V}_\xi$ is negative (it is minus the 'flat' volume excluded by the black hole, i.e. $\bar{V}_\xi = -V_{r_{\mathcal{H}}}^{\text{flat}}$), whereas for causal diamonds $V_\zeta := \int_\Sigma |\zeta| dV$ is positive.

---

[9] In the literature $\bar{V}_\xi$ is usually denoted by $\Theta$. Moreover, the background subtracted thermodynamic volume is expressed in [33] in terms of surface integrals of the Killing potential $(d-2)$-form $\omega_\xi$, defined through $\xi \cdot \epsilon = d\omega_\xi$, which can be solved at least locally for $\omega_\xi$ because $\xi \cdot \epsilon$ is closed for Killing vectors. Thus, $\bar{V}_\xi = \oint_\infty (\omega_\xi - \omega_\xi^{\text{AdS}}) - \oint_{\mathcal{H}} \omega_\xi$, where the orientation is outward (toward larger radius) at both $\infty$ and $\mathcal{H}$. This agrees with the expression (22) in [33], up to a minus sign in the definition of $\omega_\xi$.

## 3.2 First law of causal diamonds

From the Smarr formula one can derive a variational identity analogous to the first law for black holes using a simple scaling argument (see e.g. [33]). In fact, Smarr [49] originally derived the relation (3.5) for stationary black holes from the first law of black hole mechanics by using Euler's theorem for homogeneous functions, applied to the black hole mass considered as a function of the horizon area, angular momentum, and charge. In the context of causal diamonds, the area is a function of the volume and the cosmological constant alone, $A(V, \Lambda)$, since $V$ and $\Lambda$ determine a unique diamond up to isometries. It follows from dimensional analysis that $\lambda^{d-2}A(V, \Lambda) = A(\lambda^{d-1}V, \lambda^{-2}\Lambda)$, where $\lambda$ is a dimensionless scaling parameter. For a function with this property Euler's theorem implies

$$(d-2)A = (d-1)\left(\frac{\partial A}{\partial V}\right)_\Lambda V - 2\left(\frac{\partial A}{\partial \Lambda}\right)_V \Lambda. \tag{3.20}$$

Comparing this with the Smarr formula (3.16) we find that

$$\left(\frac{\partial A}{\partial V}\right)_\Lambda = k, \qquad \left(\frac{\partial A}{\partial \Lambda}\right)_V = -\frac{V_\zeta}{\kappa}, \tag{3.21}$$

which yields the first law for causal diamonds in (A)dS,

$$\kappa\, \delta A = \kappa k\, \delta V - V_\zeta\, \delta\Lambda. \tag{3.22}$$

Notice that an increase of the cosmological constant at fixed volume leads to a decrease of the area. This is because the spatial curvature is increased inside the ball. The fact that the coefficient of $\delta V$ is the extrinsic curvature $k$ can be understood by considering a variation of the radius of a ball in a fixed, maximally symmetric space. If the proper radius increase is $\delta\ell$, the volume increase is $\delta V = A\delta\ell$, while the area increase is $\delta A = kA\delta\ell$, hence $\delta A = k\delta V$.

The first law (3.21) involves only variations of the parameters that characterize the maximally symmetric causal diamond, and matter fields are not included in this approach because there are no maximally symmetric solutions with matter (except the cosmological constant). The first law can be extended to allow for variations away from maximal symmetry, thereby permitting variations of the matter stress tensor, as has been done both for black holes [2] and for de Sitter space [5]. We next derive such an extended first law by varying the identity (3.4), as was done for vacuum black holes in [26], but including matter stress-tensor variations as in Refs. [30,51]. The variations we consider are arbitrary variations of the dynamical fields $\phi$ to nearby solutions, while keeping the manifold, the vector field $\zeta^a$ and the surface $\Sigma$ of the unperturbed diamond fixed.

The variation of the Noether current (3.2) is given on shell by

$$\delta j_\chi = \omega(\phi, \delta\phi, \mathcal{L}_\chi \phi) + d(\chi \cdot \theta(\phi, \delta\phi)), \tag{3.23}$$

where $\omega(\phi, \delta_1\phi, \delta_2\phi) := \delta_1\theta(\phi, \delta_2\phi) - \delta_2\theta(\phi, \delta_1\phi)$ is the *symplectic current* $(d-1)$-form. The variation of the integral identity (3.4) thus yields

$$\int_\Sigma \omega(\phi, \delta\phi, \mathcal{L}_\chi \phi) = \oint_{\partial\Sigma} \left[\delta Q_\chi - \chi \cdot \theta(\phi, \delta\phi)\right]. \tag{3.24}$$

This relation holds provided the background equations for all the fields and the linearized constraint equations associated with the diffeomorphism generated by $\chi$ are satisfied on the hypersurface $\Sigma$.[10] The left hand side of (3.24) is the symplectic form on the (covariant) phase

---

[10]The fact that (3.24) invokes only the (linearized) initial value constraint equations (as opposed to linearized dynamical field equations), is explained in the Appendix of [52] and Appendix B of [17].

space of solutions which, by Hamilton's equations,[11] is equal to the variation of the Hamiltonian,

$$\delta H_\chi = \int_\Sigma \omega(\phi, \delta\phi, \mathcal{L}_\chi \phi). \tag{3.25}$$

Equation (3.24) thus yields the on-shell identity relating the Hamiltonian variation to the Noether charge variation and the symplectic potential,

$$\delta H_\chi = \oint_{\partial\Sigma} \left[ \delta Q_\chi - \chi \cdot \theta(\phi, \delta\phi) \right]. \tag{3.26}$$

If $\chi$ is a true Killing vector of the background metric and matter fields, then (3.25) implies $\delta H_\chi = 0$, so the variational identity reduces to a relation between the boundary integrals. This is how the first law of black hole mechanics arises [26, 46]: taking $\Sigma$ to be a hypersurface bounded by the black hole horizon and spatial infinity, the identity relates the variation of the horizon Noether charge to the variations of total energy and angular momentum.

A special case for the first law of causal diamonds is the first law of a static patch of dS space [5], which in vacuum is just the statement that the variation of the area of the de Sitter horizon vanishes. The first law derived by Gibbons and Hawking allowed for variations in the Killing energy of matter, but matter contributions do not appear in $\delta H_\chi$ if the matter is described by fields that appear quadratically in the Lagrangian and vanish in the de Sitter background. However, for matter described by a diffeomorphism invariant fluid theory first order variations of the matter stress tensor can arise, and because the fields are potentials which do not share the background Killing symmetry enjoyed by the stress tensor, a volume contribution containing the matter Killing energy appears in the variational relation [2, 30]. In the derivation of the first law for causal diamonds below we also allow for variations of fluid matter fields.

We consider the case where the gravitational theory is general relativity, the matter sector consists of minimally coupled fluids with arbitrary equation of state, the background metric is pure dS, and the vector field $\chi$ is the conformal Killing vector $\zeta$ of a causal diamond.[12] One of the fluids describes the cosmological constant, with equation of state $p = -\rho = -\Lambda/(8\pi G)$. Since $\zeta$ is zero at the edge $\partial\Sigma$ of the diamond, the second term on the right hand side in (3.26) vanishes. The surface integral of $\delta Q_\zeta$ in this case is

$$\oint_{\partial\Sigma} \delta Q_\zeta = -\frac{\kappa}{8\pi G} \delta A, \tag{3.27}$$

where $\kappa$ is the surface gravity and $A$ is the area of the bifurcation surface $\partial\Sigma$.[13] With this result, the identity (3.26) takes the form

$$\delta H_\zeta = -\frac{\kappa}{8\pi G} \delta A. \tag{3.28}$$

For the present field content the variation of the total Hamiltonian splits into a (nonvanishing) term associated to the background metric and one associated to the matter fields

$$\delta H_\zeta = \delta H_\zeta^{\mathrm{g}} + \delta H_\zeta^{\mathrm{m}}. \tag{3.29}$$

---

[11]The variation of a Hamiltonian $H$ for a general dynamical system is related to the symplectic form $\omega$ on phase space and the flow vector field $X_H$ of the background solution via Hamilton's equations, $dH(v) = \omega(v, X_H)$, where $v$ is any tangent vector on phase space. In the present case $v$ corresponds to $\delta\phi$, $X_H$ to $\mathcal{L}_\chi \phi$, and $dH(v)$ is written as $\delta H$ [53].

[12]Many steps in the derivation below remain valid for other conformally flat solutions. First of all, causal diamonds in conformally flat spacetimes still allow for a unique conformal Killing field whose flow preserves the diamond. Moreover, the diamonds still have a reflection symmetry around $\Sigma$, so that $\mathcal{L}_\zeta g_{ab} = 0$ on $\Sigma$. However, $\dot\alpha$ might not be constant in other spacetimes, so all the equations up to (3.34) hold, but not (3.35), since the relation (2.13) for $\dot\alpha$ might be specific to maximally symmetric spacetimes.

[13]The minus sign appears for the same reason as in (3.6), which is explained in footnote 8.

In the following we will first evaluate the gravitational term and then the matter term. The result will be that the gravitational term is proportional to minus the variation of the volume of $\Sigma$, the matter term contains a term proportional to the thermodynamic volume times the variation of the cosmological constant as well as the variation of the canonical Killing energy for the other fluids.

We evaluate $\delta H^{\mathrm{g}}$ through its relation to the symplectic form (3.25). For general relativity the symplectic current takes the form [50, 54]

$$\omega(g, \delta_1 g, \delta_2 g) = \frac{1}{16\pi G} \epsilon_a P^{abcdef} \left( \delta_2 g_{bc} \nabla_d \delta_1 g_{ef} - \delta_1 g_{bc} \nabla_d \delta_2 g_{ef} \right), \tag{3.30}$$

with

$$P^{abcdef} = g^{ae} g^{bf} g^{cd} - \frac{1}{2} g^{ad} g^{be} g^{cf} - \frac{1}{2} g^{ab} g^{cd} g^{ef} - \frac{1}{2} g^{bc} g^{ae} g^{fd} + \frac{1}{2} g^{bc} g^{ad} g^{ef}. \tag{3.31}$$

Note that in (3.25) the symplectic current is evaluated on the Lie derivative of the fields along $\zeta$. If $\zeta$ were a Killing vector, the metric contribution $\delta H_{\zeta}^{\mathrm{g}}$ would hence vanish, as it does when deriving the first law of black hole mechanics [26]. However, since for a diamond $\zeta$ is only a *conformal* Killing vector, $\delta H_{\zeta}^{\mathrm{g}}$ makes a nonzero contribution to the first law. When $\delta_1 g = \delta g$ and $\delta_2 g = \mathcal{L}_{\zeta} g$, the first term in (3.30) is zero at $\Sigma$, since $\mathcal{L}_{\zeta} g|_{\Sigma} = 0$ (2.8). Using (2.9) the second term yields

$$\omega(g, \delta g, \mathcal{L}_{\zeta} g)\big|_{\Sigma} = -\frac{(d-2)\dot{\alpha}}{16\pi G} \epsilon_a \left( h^{ab} u^c - h^{bc} u^a \right) \delta g_{bc}, \tag{3.32}$$

where $h_{ab} := g_{ab} + u_a u_b$ is the induced metric on $\Sigma$ and $u_a$ is the unit normal to $\Sigma$. Only the pullback of $\omega$ to $\Sigma$ is relevant in the integral in (3.25). Using

$$\epsilon_a\big|_{\Sigma} = -u_a (u \cdot \epsilon), \tag{3.33}$$

this pullback can be simplified as

$$\omega(g, \delta g, \mathcal{L}_{\zeta} g)\big|_{\Sigma} = \frac{(d-2)\dot{\alpha}}{16\pi G} (u \cdot \epsilon) h^{bc} \delta h_{bc} = \frac{(d-2)\dot{\alpha}}{8\pi G} \delta(u \cdot \epsilon), \tag{3.34}$$

where pullback of all forms to $\Sigma$ is implicit. The metric contribution to $\delta H_{\zeta}$ is therefore equal to

$$\delta H_{\zeta}^{\mathrm{g}} = \frac{d-2}{8\pi G} \int_{\Sigma} \dot{\alpha} \, \delta(u \cdot \epsilon) = -\frac{\kappa k}{8\pi G} \delta V, \tag{3.35}$$

where $V = \int_{\Sigma} u \cdot \epsilon$ is the proper volume of $\Sigma$, and in the last equality we used (2.13) and the fact that $\dot{\alpha}$ is constant over $\Sigma$. The constancy of $\dot{\alpha}$ is hence crucial for arriving at an intrinsic geometric quantity, the variation of the proper volume.

Combining (3.28), (3.29) and (3.35), we find the extended first law for causal diamonds, which includes a variation of the matter Hamiltonian,

$$\delta H_{\zeta}^{\mathrm{m}} = \frac{\kappa}{8\pi G} \left( -\delta A + k \delta V \right). \tag{3.36}$$

Next, we compute the matter Hamiltonian variation explicitly through its relation with the symplectic form.

The precise on-shell relation between the symplectic current $\omega^{\mathrm{m}}$ and the Noether current $j_{\chi}^{\mathrm{m}}$ for matter fields is [30]

$$\omega^{\mathrm{m}}(\phi, \delta\phi, \mathcal{L}_{\chi}\phi) = \delta j_{\chi}^{\mathrm{m}} + \frac{1}{2} \chi \cdot \epsilon \, T^{ab} \delta g_{ab} - d(\chi \cdot \theta^{\mathrm{m}}(\phi, \delta\phi)). \tag{3.37}$$

Here, $T^{ab}$ is the Hilbert stress-energy tensor defined through the matter Lagrangian.[14] Compared to the equivalent identity (3.23) for all the dynamical fields, we see that in the identity above for the matter sector the term involving the stress-energy tensor is new. This term arises from the metric variation of the matter Lagrangian. A similar identity exists for the pure metric sector, with the extra term being $-(1/16\pi G)\chi \cdot \epsilon\, G^{ab}\delta g_{ab}$, so that (3.23) holds when the pure metric and matter sectors are combined and the metric equation of motion is imposed.

Further, the Noether current for matter fields is on shell given by [30]

$$j_\chi^{\mathrm{m}} = dQ_\chi^{\mathrm{m}} - T_a{}^b \chi^a \epsilon_b. \tag{3.38}$$

The stress-energy term appears on the right hand side because only the full Noether current $j_\chi = j_\chi^{\mathrm{g}} + j_\chi^{\mathrm{m}}$ is an exact form on shell. Inserting (3.38) into the variational identity (3.37) and using Hamilton's equations (3.25), we find that the matter Hamiltonian variation is

$$\delta H_\chi^{\mathrm{m}} = \oint_{\partial\Sigma}\left[\delta Q_\chi^{\mathrm{m}} - \chi\cdot\theta^{\mathrm{m}}(\phi,\delta\phi)\right] + \int_\Sigma\left[-\delta(T_a{}^b\chi^a\epsilon_b) + \frac{1}{2}\chi\cdot\epsilon\, T^{ab}\delta g_{ab}\right]. \tag{3.39}$$

This equality is true for an arbitrary smooth vector field $\chi$ on spacetime, and holds provided the field equations and the linearized equations of motion are satisfied for the matter fields, i.e. $E^{\mathrm{m}} = \delta E^{\mathrm{m}} = 0$.

We now specialize to the conformal Killing vector $\zeta$ that preserves a diamond in (A)dS (the analysis below is actually valid in any conformally flat spacetime). Since $\zeta = 0$ at $\partial\Sigma$, the second term in the boundary integral in (3.39) vanishes. In addition, the Noether charge variation also does not contribute at the bifurcation surface $\partial\Sigma$. This is because for a generic diffeomorphism invariant Lagrangian the Noether charge $(d-2)$-form can be expressed as $Q_\zeta = W_c(\phi)\zeta^c + X^{cd}(\phi)\nabla_{[c}\zeta_{d]}$ [46].[15] The first term vanishes at $\partial\Sigma$, and the second term involves a form $X^{cd}$, which is purely constructed from derivatives of the Lagrangian with respect to the Riemann tensor (and its covariant derivatives). For minimally coupled matter fields, this form does not receive contributions from the matter sector, so $Q_\zeta^{\mathrm{m}} = 0$ at $\partial\Sigma$ for the present field content (and also $\delta Q_\zeta^{\mathrm{m}}$ vanishes at $\partial\Sigma$).

The matter Hamiltonian variation on the maximal slice $\Sigma$ in a diamond is therefore given by

$$\delta H_\zeta^{\mathrm{m}} = \int_\Sigma\left[-\delta(T_a{}^b\zeta^a\epsilon_b) + \frac{1}{2}\zeta\cdot\epsilon\, T^{ab}\delta g_{ab}\right], \tag{3.40}$$

which can be rewritten, using $\delta\epsilon_b = \frac{1}{2}\epsilon_b g^{cd}\delta g_{cd}$, as a sum of stress tensor and metric variation terms,

$$\delta H_\zeta^{\mathrm{m}} = \int_\Sigma\left[-\delta T_a{}^b + \frac{1}{2}(\delta_a{}^b T^{cd} - T_a{}^b g^{cd})\delta g_{cd}\right]\zeta^a\epsilon_b. \tag{3.41}$$

Notice that the trace part drops out of the second term.[16] In a maximally symmetric background the tracefree part of the stress tensor must vanish, hence the matter Hamiltonian variation takes the form

$$\delta H_\zeta^{\mathrm{m}} = -\int_\Sigma \delta T_a{}^b\zeta^a\epsilon_b, \tag{3.42}$$

---

[14]In particular, the variation of the matter Lagrangian $d$-form with respect to the matter fields $\psi$ and the metric $g_{ab}$ is given by: $\delta L^{\mathrm{m}} = E^{\mathrm{m}}\delta\psi + \epsilon\frac{1}{2}T^{ab}\delta g_{ab} + d\theta^{\mathrm{m}}(\phi,\delta\phi)$, where $E^{\mathrm{m}} = 0$ are the matter equations of motion, $T^{ab}$ is the stress-energy tensor, and $\theta^{\mathrm{m}}$ is the symplectic potential associated to $L^{\mathrm{m}}$ [30].

[15]Here we have fixed the ambiguity in the definition of the Noether charge, coming from the freedom to shift the symplectic potential $\theta$ by an exact form $dY(\phi,\delta\phi)$, such that $Y(\phi,\mathcal{L}_\zeta\phi) = 0$. If one were to allow for a nonzero $Y$ form, then the first law would not be modified, since the Noether charge variation (together with symplectic potential) associated to matter fields in (3.39) cancels anyway in the variational identity (3.26) against an identical term on right hand side. The cancellation of $\oint_{\partial\Sigma}[\delta Q_\chi^{\mathrm{m}} - \chi\cdot\theta^{\mathrm{m}}(\phi,\delta\phi)]$ in the first law was pointed out by Iyer in [30].

[16]We should have been able to anticipate this feature, but have not yet found a way to do so.

which receives contributions from all types of matter.[17]

Since the cosmological constant can be obtained from a field or fields covariantly coupled to the metric, its variation falls within the class of "matter" variations to which the first law (3.36) applies, and so may be included in (3.42). To separate out this contribution, we split the matter stress tensor as

$$T_{ab} = T_{ab}^{\tilde{m}} + T_{ab}^{\Lambda}, \tag{3.43}$$

where $T_{ab}^{\tilde{m}}$ is the stress-energy tensor of matter other than the cosmological constant, and $T_{ab}^{\Lambda} = -(\Lambda/8\pi G)g_{ab}$ is the "vacuum" energy-momentum tensor corresponding to the cosmological constant. The contribution of the $\Lambda$ term to the variation of the Hamiltonian is thus

$$\delta H_{\zeta}^{\Lambda} = \frac{V_{\zeta}\,\delta\Lambda}{8\pi G}, \tag{3.44}$$

where $V_{\zeta}$ is the thermodynamic volume defined in (3.13). Note that $V_{\zeta}$ is not varied, since the metric variation was already separated out in (3.41).

In conclusion, by inserting $\delta H_{\zeta}^{m} = \delta H_{\zeta}^{\tilde{m}} + \delta H_{\zeta}^{\Lambda}$ and (3.44) into (3.36), we arrive at the final form of the first law of causal diamonds,

$$\delta H_{\zeta}^{\tilde{m}} = \frac{1}{8\pi G}\left(-\kappa\,\delta A + \kappa k\,\delta V - V_{\zeta}\,\delta\Lambda\right). \tag{3.45}$$

We remind the reader of what all these symbols represent: $H_{\zeta}^{\tilde{m}}$ is the conformal Killing energy of matter other than the cosmological constant $\Lambda$, $\kappa$ is the surface gravity, $A$ is the area of the edge $\partial\Sigma$, $k$ is the trace of the (outward) extrinsic curvature of $\partial\Sigma$ as embedded in the maximal slice $\Sigma$, $V$ is the proper volume of the maximal slice $\Sigma$, and $V_{\zeta}$ is the proper volume weighted locally by the norm of the conformal Killing vector $\zeta$.

The derivation above also goes through for causal diamonds in AdS. Note that the form of the first law is the same for (A)dS as for Minkowski space, except that all the quantities should now be evaluated in (A)dS. Hence, we have established a variational identity in general relativity which holds for spherical regions of any size in maximally symmetric spacetimes.

### 3.3 Further remarks on the first law

Below we collect several comments on aspects of the first law of causal diamonds.

### 3.3.1 Role of maximal volume

When we evaluated the variation of the gravitational Hamiltonian $\delta H_{\zeta}^{g}$ in (3.35), we chose to carry out the integral over the maximal slice of the unperturbed diamond. Because the symplectic current is conserved, the value would have been the same had we chosen any other slice bounded by $\partial\Sigma$, although it would not have been given in the same way by the volume variation. The slice $\Sigma$ therefore has a somewhat preferred status. Furthermore, although we described this as the variation of the volume of "the slice that was the maximal slice in the unperturbed diamond," we could just as well describe it as the variation of the volume of the maximal slice itself. This is because the volume change due to the variation of the location of the maximal slice itself vanishes, precisely because that slice is maximal to begin with. This is satisfying, since it allows the first law to be stated in a manifestly "gauge-invariant" fashion — i.e. independently of how the spacetime interior of the varied diamond is identified with that of the original diamond — and the second and higher order variations of the maximal slice are unambiguously defined.

---

[17]Using (3.33), the integrand becomes $\delta T_a{}^b \zeta^a u_b\, u \cdot \epsilon$, and the unfamiliar minus sign in (3.42) disappears.

### 3.3.2 Fixed volume and fixed area variations

For variations that fix the volume, (3.36) becomes a relation between the area variation at fixed volume and the variation of the matter Hamiltonian,

$$\frac{\kappa}{8\pi G}\delta A\big|_V = -\delta H^{\mathrm{m}}_\zeta. \tag{3.46}$$

That is, the presence of positive conformal Killing energy matter produces an area deficit at fixed volume. Similarly, for variations in which the area is fixed we obtain the relation

$$\frac{\kappa k}{8\pi G}\delta V\big|_A = \delta H^{\mathrm{m}}_\zeta. \tag{3.47}$$

Hence, the presence of positive conformal Killing energy matter produces a volume excess at fixed area.

Importantly, the combination of variations $\delta A - k\delta V$ that appears in the first law (3.36) is *equivalent* to the area variation at fixed volume, i.e.

$$\delta A - k\delta V \sim \delta A\big|_V, \tag{3.48}$$

where the equivalence means modulo diffeo-induced variations. This is because (i) it is always possible to compose any variation with a variation $\delta_\xi$ induced by a diffeomorphism $\xi$, such that the volume $V$ is unchanged under the complete variation; and (ii) for *all* diffeo-induced variations the combination $\delta_\xi A - k\delta_\xi V$ vanishes (as shown below). The combination of variations $\delta A - k\delta V$ is thus equal to the area change that would remain if one were to compose a generic variation with a diffeo-variation restoring the volume to its original value.

Further, the matter Hamiltonian variation $\delta H^{\mathrm{m}}_\zeta = \delta H^{\tilde{\mathrm{m}}}_\zeta + \delta H^\Lambda_\zeta$ in the first law (3.36) is also unaffected by composing the variation with a diffeo-induced variation: since $\Lambda$ is constant in the background, $\delta_\xi \Lambda := \mathcal{L}_\xi \Lambda = 0$, and since matter (other than the $\Lambda$ contribution) can be present only after the field variation away from maximal symmetry, $\hat{\delta} H^{\tilde{\mathrm{m}}}_\zeta$ is non-vanishing only at the next variational order. Thus, both sides of the first law vanish for variations that are induced by diffeomorphisms. Hence, we are free to add to the first law (3.36) a diffeo-induced variation that restores the volume $V$ to its original value, so that the first law takes the form (3.46). Similarly, one can also freely add a diffeo-induced variation that restores the area $A$ to its original value, such that to the first law becomes (3.47). This means that the first law at fixed volume (3.46) and the first law at fixed area (3.47) are equivalent to the one (3.36) without holding the volume and area fixed.

Finally, we prove statement (ii) above. That is, if $\mathcal{R}$ is a codimension-one submanifold with vanishing mean extrinsic curvature, and if the boundary $\partial\mathcal{R}$ has mean extrinsic curvature (normal to $\partial\mathcal{R}$) that vanishes in the direction normal to $\mathcal{R}$ and is constant in the direction tangent to $\mathcal{R}$, then under an infinitesimal diffeo-variation the variations of the area $A$ of $\partial\mathcal{R}$ and the volume $V$ of $\mathcal{R}$ are related by[18]

$$\delta_\xi A - k\,\delta_\xi V = 0, \tag{3.49}$$

---

[18]If the assumptions are relaxed, then the diffeo-variation of the volume is

$$\delta_\xi V = \int_{\mathcal{R}} K\sigma_a u^a u \cdot \epsilon + \int_{\partial\mathcal{R}} \left( \frac{1}{k_n}\delta_\tau \mu + \frac{k_u}{k_n}\sigma_a u^a \mu \right).$$

Here, $\sigma$ and $\tau$ are the components of $\xi$ normal and tangent to $\mathcal{R}$, respectively, $\mu$ is the volume form on the codimension-two surface $\partial\mathcal{R}$, and $k_u$ and $k_n$ are the mean curvatures in the two normal directions to $\partial R$. Under the assumptions $K = 0 = k_u$ and $k_n$ is constant, we recover (3.49) with $k := k_n$. A similar expression for the volume variation is given by equation (4.32) in [55]. We thank Antony Speranza for clarifying the required assumptions for the validity of (3.49).

where $k$ is the mean curvature of $\partial\mathcal{R}$ in $\mathcal{R}$. To see this, consider the diffeomorphism as an active transformation of the spacetime points. Under the stated assumptions, the normal component of the diffeomorphism will do nothing to $V$ and $A$, while the piece tangential to $\mathcal{R}$ will deform each surface element $dA$ by $\delta_\xi dA = k\, dA\, \delta_\xi s = k\, \delta_\xi dV$, where $\delta_\xi s$ is the normal deformation distance. If $k$ is constant on $\partial\mathcal{R}$, then when integrated over $\partial\mathcal{R}$ this yields $\delta_\xi A = k\delta_\xi V$. The maximal slice $\Sigma$ of a maximally symmetric diamond possesses the assumed properties, so the result applies in particular to that surface.

### 3.3.3 Varying the cosmological constant

In a thermodynamic interpretation, causal diamonds in maximally symmetric spacetimes are all "equilibrium states" from which variations can be made. The diamonds differ only in size, and in the cosmological constant of the background. It is natural to allow also $\Lambda$ to vary, since it is evidently an equilibrium state variable, and there are circumstances under which it might vary. For instance, there may be mechanisms by which it can decay. Also, in the context of the AdS/CFT correspondence, the negative cosmological constant is controlled by the number of stacked D-branes, which could in principle change [33]. Another reason to consider variable $\Lambda$ arises in formulating the principle of vacuum entanglement equilbrium for non-conformal matter fields, see Sec. 4.2.2 and Ref. [21]. Consistency with the Bianchi identity made it necessary to allow for an initially undetermined local cosmological constant in small causal diamonds, which ended up being related to the part of the entanglement entropy variation not captured by the energy-momentum tensor. It is thus of interest to include variations of $\Lambda$ in the first law. Ref. [56] provides an extensive review of black hole thermodynamics extended to include variable $\Lambda$, a.k.a. "black hole chemistry".

There are many ways to accommodate a cosmological constant variation in the first law. In the literature this has been done for the first law for black holes and for holographic entanglement entropy by employing various methods, see e.g. [33,56–60]. In Sec. 3.2 we treated the cosmological constant as a perfect fluid, and made use of Iyer's generalized derivation of the first law to allow for matter fields which are non-stationary yet have a stationary stress-energy tensor [30]. In this approach the cosmological constant term in the first law comes from the variation of the stress-energy tensor of the fluid. Yet another way of introducing a cosmological constant is to promote it to a dynamical scalar field, and to add it to the Lagrangian together with a $(d-1)$-form field $B$ as: $\Lambda(dB-\epsilon)/(8\pi G)$ [61]. The $B$ field equation implies that $\Lambda$ is constant, while the $\Lambda$ field equation implies $dB = \epsilon$. The addition to the symplectic potential due to this Lagrangian is $\theta(\phi,\delta\phi) = \Lambda\delta B/(8\pi G)$, where $\phi = (\Lambda, B)$. Moreover, the additional term in the symplectic current is given on shell by $\omega(\phi,\delta\phi,\mathcal{L}_\zeta\phi) = [\delta\Lambda\,\zeta\cdot\epsilon + d(\delta\Lambda\,\zeta\cdot B)]/(8\pi G)$. When integrated over $\Sigma$ this gives precisely $V_\zeta\delta\Lambda/(8\pi G)$, as in (3.44), since $\zeta$ vanishes at the edge $\partial\Sigma$.

### 3.3.4 Gravitational field Hamiltonian and York time

We have seen that the gravitational contribution to the variation of the Hamiltonian $H_\zeta$ generating evolution along the conformal Killing flow of the background maximally symmetric diamond is proportional to the volume variation. This "volume as Hamiltonian" is reminiscent of a "York time" Hamiltonian for general relativity [36], which generates evolution along a foliation by spacelike hypersurfaces with constant mean curvature $K$ (i.e. along a "CMC" foliation), using $K$ as the time parameter, and with an arbitrary shift vector field. (Mean curvature can be defined as $K := \nabla_a u^a$, where $u^a$ is the future pointing unit normal to a spacelike hypersurface.) Such a Hamiltonian is proportional to the spatial volume of the CMC slices.

The similarity is not accidental. It arises from the fact that (i) the conformal Killing vector $\zeta^a$ is orthogonal to $\Sigma$, which is a CMC surface, and (ii) $\zeta^a\nabla_a K$ is constant on $\Sigma$. Actually,

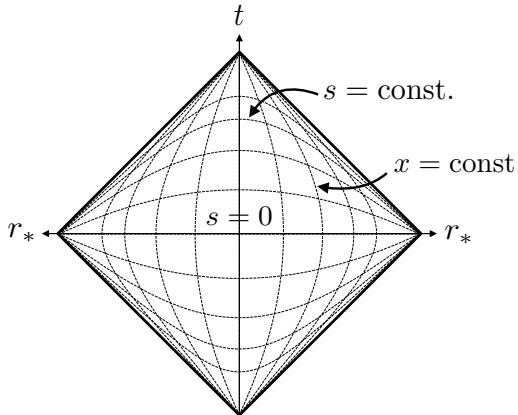

Figure 2: The $(x, s)$ coordinate chart of a maximally symmetric causal diamond. The coordinate $s \in (-\infty, \infty)$ is the conformal Killing time, defined as the function that vanishes on the maximal slice $\Sigma$ and satisfies $\zeta \cdot ds = 1$. The coordinate $x \in [0, \infty)$ is spherically symmetric and satisfies $\zeta \cdot dx = 0$ and $|dx| = |ds|$. Constant $s$ and $x$ lines are plotted at equal coordinate intervals of 0.5. See Appendix B for a demonstration that $ds$ and $dx$ are everywhere orthogonal, and for the line element in these coordinates.

these two properties hold on *all* leaves of the CMC foliation: as shown in Appendix B, surfaces of constant conformal Killing parameter $s$ — defined by $\zeta^a \nabla_a s = 1$ with the initial condition $s = 0$ on $\Sigma$ — coincide with surfaces of constant $K$ everywhere in the diamond, and $\zeta^a$ is everywhere orthogonal to these surfaces (see Fig. 2 for an illustration). More specifically, $K$ and $s$ are related by

$$K = (d-1)\dot\alpha|_{s=0} \sinh s, \tag{3.50}$$

where $\alpha = \nabla_a \zeta^a / d$ and $\dot\alpha = u^a \nabla_a \alpha$. In particular, $K$ vanishes at the extremal surface $s = 0$, and its first derivative with respect to $s$ at $s = 0$ is given by

$$\left.\frac{dK}{ds}\right|_{s=0} = (d-1)\dot\alpha|_{s=0} = -\frac{d-1}{d-2}\kappa k, \tag{3.51}$$

where (2.13) is used in the last equality.[19] Equation (3.51) establishes that York time and conformal Killing time are proportional, to first order about the maximal slice, for a maximally symmetric diamond. This indicates, as we will now argue, that the variation $\delta H_\zeta^g$ agrees, up to the constant (3.51), with the York time Hamiltonian variation $\delta H_Y$.

In the context of the first law, the perturbed spacetime is not the maximally symmetric one. When the metric is varied, the definition of York time varies, so the surface on which we should be computing the volume varies, as does the rate of time flow. Nevertheless, since the $t = 0$ surface has vanishing $K$, the volume variation induced by varying the surface vanishes. Also, the field variation is already first order, so the change of flow rate of $K$ makes a higher order contribution to $\delta H_Y$. It follows that

$$\delta H_\zeta^g = \left.\frac{dK}{ds}\right|_{s=0} \delta H_Y, \qquad \text{with} \qquad \delta H_Y = \frac{d-2}{d-1}\frac{\delta V}{8\pi G}. \tag{3.52}$$

Therefore, the gravitational Hamiltonian variation (3.35) is indeed equal to a constant times the York time Hamiltonian variation. The York time Hamiltonian variation would be equal to the negative of the proper volume variation, i.e. $\delta H_Y = -\delta V$, if one used the time variable $t_Y = -\frac{d-2}{d-1}\frac{K}{8\pi G}$ instead of $K$. This is precisely the time variable that York originally introduced for general relativity in $d = 4$ [36]. In the literature the minus sign in the time variable is often omitted, in which case the Hamiltonian is equal to the volume (see e.g. [62]).

---

[19]Note that $K$ decreases as $s$ increases, and is hence negative to the future of the slice $\Sigma$ (and positive to the past of $\Sigma$).

# 4 Thermodynamics of causal diamonds in (A)dS

As is well known black holes admit a true thermodynamic interpretation. In this section we will explore to what extent the same is true for causal diamonds in (A)dS. We will also relate the first law of causal diamonds to the entanglement equilibrium proposal in [21].

## 4.1 Negative temperature

Like the first law of black hole mechanics, and its generalizations mentioned in the introduction, the first law of causal diamonds (3.28),

$$\delta H_\zeta = -\frac{\kappa}{8\pi G}\delta A, \tag{4.1}$$

admits a thermodynamic interpretation. What is unusual, however, is the minus sign on the right hand side. The $\kappa\,\delta A$ term is usually identified with a $T_{\mathrm{H}}\delta S_{\mathrm{BH}}$ term, where $S_{\mathrm{BH}} = A/4\hbar G$ is the Bekenstein-Hawking entropy and $T_{\mathrm{H}} = \kappa\hbar/2\pi$ is the Hawking temperature. However, in the present context this identification calls for a negative temperature[20]

$$T = -T_{\mathrm{H}}, \tag{4.2}$$

because an increase of conformal Killing energy in the diamond is associated with a *decrease* of horizon entropy.[21] This negative temperature interpretation has previously been suggested by Klemm and Vanzo [31] in the special case of the static patch of de Sitter space, and was recently advocated in the context of multiple Killing horizons in [32], where the cosmological event horizon was also assigned a negative (Gibbsian) temperature.[22] Negative temperature typically requires of a system that i) its energy spectrum is bounded above and ii) the Hilbert space is finite-dimensional. Klemm and Vanzo have argued that these requirements are indeed satisfied for the de Sitter space static patch.[23] Their arguments can actually be applied to all causal diamonds: i) the mass inside is bounded above by the mass of the largest black hole that fits inside a diamond with a given boundary area and ii) the entropy associated to the horizon is finite due to the holographic principle or covariant entropy bound [71,72]. It therefore seems feasible that causal diamonds have a negative temperature in quantum gravity.

Using the negative temperature (4.2), the first law (4.1) can be written as

$$\delta H_\zeta = T\delta S_{\mathrm{BH}}, \tag{4.3}$$

---

[20]The conceptual possibility of negative absolute temperature was discussed for the first time by Afanassjewa in 1925 [63]. In 1951 Purcell and Pound [64] prepared and measured a nuclear spin system at negative temperature in an external magnetic field. Subsequently, the thermodynamic and statistical mechanical implications of negative temperature were studied in detail by Ramsey [65]. We thank Jos Uffink for bringing the work of Afanassjewa to our attention [66]. See [67] for a recent review of negative temperature.

[21]It was suggested in Ref. [68] that in global de Sitter space this minus sign (which was there attached to the entropy rather than to the temperature) results from imposing the first law with the energy inside the horizon, rather than the energy outside the horizon which is the negative of the former. The sensibility of this proposal is debatable, since the opposite sign of the energy results from the fact that the Killing vector is past-oriented outside the horizon. Moreover, in flat spacetime or AdS the energy outside the horizon is not the negative of that inside, and yet we still encounter the same minus sign.

[22]The arguments in [32] appealed to the fact that the surface gravity of the cosmological horizon is negative. Although not stated in [32] (nor elsewhere in the literature that we are aware of), this holds only on the *past* cosmological horizon (if we take the Killing vector to be future pointing on the past and future horizons). The surface gravity of the *future* cosmological horizon is *positive*. (The surface gravity is positive (negative) if the Killing vector is stretched (shrunk) with respect to affine parameter along the Killing flow on the Killing horizon.) A varied diamond can be viewed as the result of a physical process in which a perturbation has passed through the past horizon, and entered what would otherwise have been a maximally symmetric diamond. The surface gravity of the *past* horizon should thus be expected to play the role of temperature in the first law.

[23]The proposal that the Hilbert space of an observer's patch in asymptotic de Sitter space is finite-dimensional is due to Banks and Fischler [69,70].

which is a standard thermodynamic relation between energy, temperature and entropy. As a special case, we find that the static patch of de Sitter space has a negative temperature (see Sec. 5.1). This is in apparent conflict with the positive Gibbons-Hawking temperature for dS space, computed using quantum field theory on a fixed background [5]. In Sec. 4.2 below we shall propose a resolution to this apparent conflict involving the quantum corrections to the first law, but first we want to further discuss the thermodynamic interpretation of the leading order classical quantities in this relation.

Instead of writing the cosmological constant term (3.44) in $\delta H_\zeta$ as an energy variation, we can also take it to the right hand side of the first law and write it as the thermodynamic volume times the variation of the pressure, i.e. $V_\zeta \delta p$. This is because the cosmological constant can be interpreted both as an energy density, $\rho = \Lambda/8\pi G$, and as a pressure $p = -\Lambda/8\pi G$. In this way (4.3) can be expressed as

$$\delta H_\zeta^{\mathrm{g+\tilde{m}}} = T\delta S_{\mathrm{BH}} + V_\zeta \,\delta p \,, \tag{4.4}$$

where g labels the gravitational contribution (1.4) to the Hamiltonian variation, and $\tilde{\mathrm{m}}$ labels the matter contribution (1.2) other than the cosmological constant. This form of the first law suggests that $H_\zeta^{\mathrm{g+\tilde{m}}}$ is an enthalpy, rather than an energy, just like the ADM mass for black holes [33]. The matter Hamiltonian vanishes in the background, so the enthalpy of causal diamonds in Minkowski and (A)dS space is $H_\zeta^{\mathrm{g}}$, which is defined above only through its variation. We leave it to future work to evaluate $H_\zeta^{\mathrm{g}}$ itself.

Through a Legendre transformation, $U = H_\zeta^{\mathrm{g+\tilde{m}}} - p\,V_\zeta$, the first law can be written in the standard form

$$\delta U = T\delta S_{\mathrm{BH}} - p\,\delta V_\zeta \,, \tag{4.5}$$

where $U$ plays the role of the internal energy associated to causal diamonds. Using the equation of state $p = -\rho$, the contribution of the $\Lambda$ term to $U$ may be expressed as $\rho\,V_\zeta$, which is the redshifted vacuum energy associated to the cosmological constant. (See Sec. 5.1 for a similar discussion for the special case of the de Sitter static patch.)

## 4.2 Quantum corrections

The first law can be extended into the semiclassical regime by considering quantum matter fields (instead of classical fields) on a classical background spacetime. The "quantum corrected" first law of causal diamonds reads

$$\delta \langle H_\zeta^{\tilde{\mathrm{m}}} \rangle + \delta H_\zeta^{\mathrm{g+\Lambda}} = T\delta S_{\mathrm{BH}} \,, \tag{4.6}$$

which can be derived (along the lines of Sec. 3.2) from the semiclassical Einstein equation, where the stress-energy tensor is replaced by its quantum expectation value but the metric is kept classical. Our aim in this section is to show how this first law can be written in terms of the variation of Bekenstein's generalized entropy [1] — defined in (4.11) — which at the same time explains why the negative temperature $T$ is consistent with the positive Gibbons-Hawking temperature $T_{\mathrm{H}}$. We will first restrict the discussion to conformal matter and then generalize it to any quantum matter.

### 4.2.1 Conformal matter

In the particular case of the vacuum state of a conformal quantum field theory, the matter Hamiltonian $H_\zeta^{\tilde{\mathrm{m}}}$ associated to a spherical region $\Sigma$ in flat space or (A)dS is equal to the so-called modular Hamiltonian $K$, i.e.

$$H_\zeta^{\tilde{\mathrm{m}}} := \int_\Sigma (T_a{}^b)^{\tilde{\mathrm{m}}} \zeta^a u_b dV = K \,, \tag{4.7}$$

where $K$ is the operator implicitly defined via the reduced density matrix of the vacuum restricted to the region $\Sigma$, $\rho_{\text{vac}} = e^{-K/T_H}/Z$.[24] For infinitesimal variations of the reduced density matrix, the variation of the expectation value of the modular Hamiltonian is equal to the variation of the matter entropy $S_{\tilde{\text{m}}}$, by

$$\delta \langle K \rangle = T_{\text{H}} \delta S_{\tilde{\text{m}}}, \tag{4.8}$$

with the *positive* sign for the temperature $T_{\text{H}}$. If the variation is to a global pure state $\delta S_{\tilde{\text{m}}}$ is purely entanglement entropy, which is why this is known as the "first law of entanglement".

Initially it appears that this opposite sign for the temperature indicates an inconsistency: if $\delta \langle H_\zeta^{\tilde{\text{m}}} \rangle$ were added to the rest of the conformal energy variation $\delta H_\zeta$, and at the same time $\delta S_{\tilde{\text{m}}}$ were added to $\delta S_{\text{BH}}$ in the classical first law (4.3), then — because of the mismatch in the signs of the temperatures — the first law would no longer be valid for any temperature. However, in gravitational thermodynamics, it would be incorrect to add both the energy term and the entropy term. The derivation of the *gravitational* first law follows from diffeomorphism invariance and the gravitational field equation, and the matter entropy does not enter in the derivation [26, 30]. On the other hand, this first law must be consistent with the *thermodynamic* first law, so it must also be possible to take into account the matter entropy.

The way this works is perhaps easiest to understand in the setting of a stationary, asymptotically flat black hole surrounded by a fluid [2, 30]. The gravitational first law indicates that a matter Killing energy variation $\delta H_\xi^{\tilde{\text{m}}}$ increases the variation of the total ADM mass $M$,

$$\delta M = \delta H_\xi^{\tilde{\text{m}}} + \kappa \delta A/8\pi G, \tag{4.9}$$

where $\xi$ is the time translation Killing vector, and variation of angular momentum of the black hole is suppressed for simplicity.[25] On the other hand, the energy variation of a thermal fluid can be expressed in terms of its *entropy* and *particle number* density variations, $\delta dS_{\tilde{\text{m}}}$ and $\delta dN$, and redshifted comoving temperature and chemical potential, $\bar{T}$ and $\bar{\mu}$, together with a possible angular momentum variation that is also suppressed. If that is done, the entropy of the fluid registers *explicitly* in the thermodynamical first law,

$$\delta M = \int_\Sigma (\bar{T} \delta dS_{\tilde{\text{m}}} + \bar{\mu} \delta dN) + \kappa \delta A/8\pi G, \tag{4.10}$$

while the *energy* of the fluid registers only *implicitly* in the $\delta M$ term, to which it contributes via the constraints. It is quite peculiar to gravitational thermodynamics that the first law has simultaneously these two different meanings, one gravitational and one thermodynamical [73], and that the fluid does not contribute explicitly to both the entropy and energy variations, unlike in ordinary thermodynamics.

It thus seems that the correct procedure is to add *only* the matter energy variation $\delta \langle H_\zeta^{\tilde{\text{m}}} \rangle$ to the classical first law, as we anticipated in (4.6), where the classical matter Hamilonian

---

[24] It is well known that the reduced density matrix of the vacuum in Rindler space is thermal with respect to the Lorentz boost Hamiltonian [23]. The thermal behavior of conformal quantum fields in a global vacuum state inside maximally symmetric causal diamonds can be derived from the Weyl equivalence between these diamonds and Rindler space. In Appendix B we show explicitly that all maximally symmetric diamonds are Weyl equivalent to conformal Killing time cross hyperbolic space, $\mathbb{R} \times \mathbb{H}^{d-1}$, and therefore to each other. It is also known that a diamond in flat space is Weyl equivalent to Rindler space, see Appendix E. Therefore, the reduced density matrix of the conformal vacuum on a diamond in (A)dS and flat space is thermal. For the special case of diamonds in flat space this was discussed before in [12, 22–25].

[25] Equation (4.9) is equivalent to the first law as given in Ref. [2, 30], although that is not immediately apparent since we write the fluid term as the variation of its contribution to the Killing Hamiltonian. The equivalence can be established using (3.39) for this variation. The $\Sigma$ integral in (3.39) is the same as the fluid contribution in Eqn. (50) of [30], and it can be shown that the $\partial\Sigma$ boundary term vanishes for the case of a prefect fluid (using the variational formulation given by Schutz and reviewed in [30]).

variation $\delta H^{\tilde{\text{m}}}$ is *replaced* by its expectation value. If desired, one can use (4.7) and (4.8) to express $\delta\langle H^{\tilde{\text{m}}}_\zeta \rangle$ in terms of the matter entropy variation. Thanks to the opposite sign of the temperature in (4.8), this can then be combined with $\delta S_{\text{BH}}$ in (4.6) to form the variation of the *generalized entropy*,

$$S_{\text{gen}} := S_{\text{BH}} + S_{\tilde{\text{m}}}, \tag{4.11}$$

in terms of which the quantum corrected first law of causal diamonds becomes

$$\delta H^{\text{g}+\Lambda}_\zeta = T\delta S_{\text{gen}}. \tag{4.12}$$

In fact, this appears to be a more satisfactory formulation of the first law because, unlike the matter entropy and Bekenstein-Hawking entropy separately, the generalized entropy is plausibly invariant under a change of the UV cutoff (see the appendix of [74] for a discussion of this idea).

In this way, we see that the opposite sign of the temperature in (4.3) and (4.8) is precisely what is needed in order for the Bekenstein-Hawking entropy and matter entropy to combine as $S_{\text{gen}}$. At least for conformal fields, this resolves the apparent conflict alluded to before between the negative temperature in the first law for causal diamonds and the positive temperature in the first law of entanglement. The negative and positive temperature seem compatible with each other, since the first law of entanglement (4.8) and the quantum corrected first law of causal diamonds (4.12) are valid simultaneously.

### 4.2.2 Non-conformal matter

We now show how the generalized entropy variation can be obtained in the first law for generic quantum fields. For non-conformal fields the matter Hamiltonian is not equal to the modular Hamiltonian, and hence the term $\delta\langle H^{\tilde{\text{m}}}_\zeta \rangle$ cannot be directly related to the entanglement entropy variation. However, in [21] it was postulated that, for causal diamonds that are small compared to the local curvature scale, the length scale of the quantum state, and any length scale in the quantum field theory defined by a relevant deformation of a conformal field theory, this term is in fact related to the variation of the expectation value of the modular Hamiltonian via

$$\delta\langle H^{\tilde{\text{m}}}_\zeta \rangle = \delta\langle K \rangle - V_\zeta\,\delta X. \tag{4.13}$$

Here $X$ is a spacetime scalar that can depend on the size of the diamond but is invariant under Lorentz boosts that leave the center of the diamond fixed. The thermodynamic volume $V_\zeta$ has been factored out for later convenience.[26] The modular Hamiltonian $K$ is here defined for the vacuum of a quantum field theory restricted to a ball-shaped region, and the variation denotes a perturbation of the vacuum state. The assumption (4.13) was checked in [37, 38], and it was found in particular that $\delta X$ may depend on the radius $R$ of the ball, and can dominate at small $R$ (depending on the conformal dimension of the operator that deforms the CFT).

With the postulate (4.13) and the first law of entanglement (4.8), the quantum first law (4.6) can be written in terms of the generalized entropy variation as

$$\delta H^{\text{g}+\lambda}_\zeta - V_\zeta\,\delta X = T\delta S_{\text{gen}}. \tag{4.14}$$

Here, we have denoted the local cosmological constant in a small maximally symmetric diamond by $\lambda$, in order to distinguish it from the total cosmological constant variation to be

---

[26]For small diamonds the conformal Killing energy variation can be approximated by $\delta\langle H^{\tilde{\text{m}}}_\zeta \rangle = V_\zeta\delta\langle T^{\tilde{\text{m}}}_{00}\rangle$ and the thermodynamic volume is to first order given by $V_\zeta = \kappa\,\Omega_{d-2}R^d/(d^2-1)$ (see Sec. 5.2). Inserting these approximations into (4.13) yields the actual conjecture (21) in [21].

introduced below. The variations $\delta H^{\mathrm{g}}$ and $\delta H^\lambda$ are explicitly given by (1.4) and (1.3), respectively, so we can further rewrite this first law as

$$-\frac{\kappa k}{8\pi G}\delta V + \frac{V_\zeta}{8\pi G}(\delta\lambda - 8\pi G\delta X) = T\delta S_{\mathrm{gen}}. \qquad (4.15)$$

Now, since the $\delta X$ contribution from non-conformal matter appears together with the local cosmological constant variation $\delta\lambda$ in this way, we may combine them into one net variation,

$$\delta\Lambda := \delta\lambda - 8\pi G\delta X, \qquad (4.16)$$

in terms of which (4.15) is expressed as

$$-\frac{\kappa k}{8\pi G}\delta V + \frac{V_\zeta}{8\pi G}\delta\Lambda = T\delta S_{\mathrm{gen}}. \qquad (4.17)$$

The first law including non-conformal quantum fields is thus also expressed in terms of the generalized entropy variation, and the temperature in this first law is still negative. We conclude that the assignment of a negative temperature to the diamond remains consistent when extended to the semiclassical realm.

### 4.3 Entanglement equilibrium

If the proper volume $V$ and cosmological constant $\Lambda$ are held fixed in (4.17), then the generalized entropy is stationary in a maximally symmetric vacuum,

$$\delta S_{\mathrm{gen}}\big|_{V,\Lambda} = \delta S_{\mathrm{BH}}\big|_{V,\Lambda} + \delta S_{\tilde{\mathrm{m}}} = 0. \qquad (4.18)$$

There is no need to fix $V$ and $\Lambda$ in the matter entropy variation, because there is no first order metric variation of the matter entropy, since the zeroth order matter state is the vacuum. The condition $\delta\Lambda = 0$ means that $\delta\lambda$ is chosen to cancel the change in the effective local cosmological constant, $-8\pi G\delta X$.

In [21], it was shown, assuming the conjecture (4.13), that the semiclassical Einstein equation holds if and only if the generalized entropy is stationary at fixed volume in small local diamonds everywhere in spacetime.[27] The validity of the latter property was called the "maximal vacuum entanglement hypothesis", but we shall refer to it as the *entanglement equilibrium* hypothesis. Here we have deduced the entanglement equilibrium statement (4.18) from the conjecture (4.13) together with the quantum corrected first law of causal diamonds (4.6), which itself was derived from the semiclassical gravitational equations of motion. In this sense the semiclassical Einstein equation is equivalent to the quantum corrected first law for small, and therefore maximally symmetric diamonds. However, the variations in the entanglement equilibrium setting [21] and in the present paper are viewed somewhat differently, so the precise relation between the two results is not immediately clear. We shall now explain how they may be brought into alignment.

In the setting of [21], an arbitrary spacetime and matter state, $(g, |\psi\rangle)$, are considered, in every small causal diamond, as a variation of a maximally symmetric spacetime (MSS) and vacuum $(g_\lambda, |0\rangle)$, with an initially arbitrary $\lambda$. The idea is that any spacetime is locally close to an "equilibrium" state, and that all maximally symmetric states qualify as equilibria. The generalized entropy $S_{\mathrm{gen}}$ in the diamond is then compared to that of the diamond with the same volume in the MSS, the difference being

$$\delta S_{\mathrm{gen}}|_{V,\lambda} = S_{\mathrm{gen}}|_{V(g,|\psi\rangle)} - S_{\mathrm{gen}}|_{V(g_\lambda,|0\rangle)}. \qquad (4.19)$$

---

[27]In that context, the area term in the generalized entropy was assumed to have the form $\eta A$, and the gravitational constant $G$ was found to be given by $1/4\hbar\eta$.

The notation suggests "fixed $\lambda$", but at this stage $\lambda$ is just an arbitrary background value for the comparison. The entanglement equilibrium assumption amounts to the postulate that there exists *some* $\lambda$, in each diamond, for which the stationary condition $\delta S_{\text{gen}}|_{V,\lambda} = 0$ holds. When applied to all diamonds this condition, together with energy-momentum conservation and the Bianchi identity, implies that $\lambda$ for each diamond is determined, up to one overall spacetime constant $\Lambda$, by the $\delta X$ of the state, and it implies that the Einstein equation holds for that $\Lambda$.

To bring this more in line with the variational relations of the present paper, instead of setting the difference (4.19) to zero, we may first reckon the diamond entropy of $(g, |\psi\rangle)$ relative to that of a diamond in flat spacetime. In the notation of [21] this yields

$$\delta S_{\text{gen}}|_V = \eta \delta A|_V + \frac{2\pi}{\hbar} \delta\langle K\rangle = \frac{\Omega_{d-2} R^d}{d^2 - 1}\left[-\eta G_{00} + \frac{2\pi}{\hbar}(\delta\langle T_{00}\rangle + \delta X)\right]. \tag{4.20}$$

Now, rather than postulating that this variation vanishes, we postulate that it is the same as would be obtained by varying from the Minkowski vacuum to a MSS vacuum with *some* cosmological constant $\lambda$ (3.22),

$$\delta_{\text{MSS}} S_{\text{gen}}|_V = \eta\, \delta_{\text{MSS}} A|_V = -\frac{\Omega_{d-2} R^d}{d^2 - 1}\eta\, \lambda. \tag{4.21}$$

The equality of (4.20) and (4.21) implies the relation

$$G_{00} + \lambda\, g_{00} = \frac{2\pi}{\hbar\eta}(\delta\langle T_{00}\rangle - \delta X\, g_{00}), \tag{4.22}$$

(since $g_{00} = -1$ in Riemann normal coordinates at the center of the diamond) and the validity of this relation for all small diamonds in spacetime implies the tensor equation

$$G_{ab} + \Lambda\, g_{ab} = \frac{2\pi}{\hbar\eta}\delta\langle T_{ab}\rangle, \tag{4.23}$$

where[28]

$$\Lambda := \lambda + \frac{2\pi}{\hbar\eta}\delta X. \tag{4.24}$$

With the identification $\eta = 1/4\hbar G$, the relation (4.24) matches that found in (4.16), when it is recognized that $\lambda$ and $\Lambda$ here refer to a maximally symmetric *comparison* spacetime, whereas in (4.16) the background spacetime is implicit and $\delta\lambda$ and $\delta\Lambda$ are part of the one overall variation that is made. Adding $\lambda$ to the comparison spacetime yields the same change of entropy as including a variation $\delta\lambda = -\lambda$ in the variation being considered, and similarly for $\Lambda$, so that the appropriate identification is $\lambda = -\delta\lambda$ and $\Lambda = -\delta\Lambda$. This establishes how the first law derived in the present paper from the Einstein equation is related to the entanglement equilibrium postulate used in [21] to derive the Einstein equation.

## 4.4 Free conformal energy

In this section we address two questions concerning the entanglement equilibrium proposal. First, in standard thermodynamics the stationarity of entropy at fixed energy follows from the stationarity of free energy at fixed temperature. Hence, it is natural to ask whether we can identify a thermodynamic potential (e.g. free energy) in our setting whose stationarity corresponds to an equilibrium condition. The stationarity is a condition on the first order variation of the free energy. Whether the free energy is minimized or maximized can only be

---

[28]Eq. (4.24) agrees with the result in Ref. [21] after correcting the sign error there of the $\delta X$ term in (25), which appears due to an error in equation (24).

determined from the second order variation of the free energy, which we leave for future work (see also [43]).[29] Second, an essential ingredient in the derivation of the Einstein equation in [21] was the fixed volume requirement. One advantage of characterizing the equilibrium state in terms of free energy stationarity, instead of entropy stationarity, is that the fixed volume constraint is relaxed. It is desirable to understand the fixed volume requirement better, and to see whether the free energy stationarity can be used as an input assumption in deriving the Einstein equation.

Let us start with the free energy for classical matter configurations. The classical first law (4.3) implies that the *free conformal energy*[30]

$$F = H_\zeta - TS_{\text{BH}} \tag{4.25}$$

is stationary at fixed temperature, i.e. $\delta F = -S_{\text{BH}}\delta T = 0$.[31] This means that causal diamonds in flat space and (A)dS are equilibrium states.

Next, in the semiclassical regime the quantum corrected free conformal energy can be defined by replacing the matter Hamiltonian by its quantum expectation value

$$F_{\text{quan}} = \langle H_\zeta^{\tilde{m}} \rangle + H_\zeta^{\text{g}+\Lambda} - TS_{\text{BH}}. \tag{4.26}$$

The stationarity of the quantum free energy at fixed temperature in the vacuum, i.e. $\delta F_{\text{quan}} = 0$, follows from the quantum first law of causal diamonds (4.6). We woud like to show that the free energy stationarity is equivalent to the entanglement equilibrium hypothesis (4.18), i.e. the stationarity of generalized entropy at fixed $V$ and $\Lambda$. In establishing the equivalence we will treat conformal quantum fields and non-conformal quantum matter separately. Using the expressions (1.3) and (1.4) for $H_\zeta^\Lambda$ and $H_\zeta^{\text{g}}$, respectively, in terms of the variation of the volume and the cosmological constant, we see that the variation of the quantum free energy (4.26) at fixed $V$ and $\Lambda$ is equal to

$$\delta F_{\text{quan}}\big|_{V,\Lambda} = \delta \langle H_\zeta^{\tilde{m}} \rangle - T\delta S_{\text{BH}}\big|_{V,\Lambda}. \tag{4.27}$$

For conformal matter, the first law of entanglement (4.8) can be used to trade the conformal Killing energy variation $\delta\langle H_\zeta^{\tilde{m}}\rangle$ for a matter entropy variation, which then combines with $\delta S_{\text{BH}}$ to form the generalized entropy variation, i.e. $\delta F_{\text{quan}}|_{V,\Lambda} = T_{\text{H}}\delta S_{\text{gen}}|_{V,\Lambda}$. The free energy for conformal quantum fields is thus stationary at fixed $V$ and $\Lambda$ if and only if the generalized entropy is stationary at fixed $V$ and $\Lambda$.

Dealing with non-conformal matter is more subtle since, to relate the variation of the conformal Killing energy to the matter entropy variation, apart from the first law of entanglement we need the additional assumption (4.13), which holds only for small diamonds. For small maximally symmetric diamonds we denote the local cosmological constant by $\lambda$, rather than $\Lambda$ in (4.27), and the variation of the associated Hamiltonian is $(V_\zeta/8\pi G)\delta\lambda$. Inserting the assumption (4.13) for $\delta\langle H_\zeta^{\tilde{m}}\rangle$ and the expression above for $\delta H_\zeta^\lambda$ into the variation of the quantum free energy at fixed volume yields

$$\delta F_{\text{quan}}\big|_V = T_{\text{H}}\delta S_{\text{gen}}\big|_V + \frac{V_\zeta}{8\pi G}(\delta\lambda - 8\pi G\delta X). \tag{4.28}$$

---

[29]We expect that maximally symmetric causal diamonds have *maximum* free energy. This is because local thermodynamic stability requires that entropy be maximized at fixed energy, which for systems with negative temperature (such as our diamonds) implies that free energy is maximized (see e.g. [75]). We thank Batoul Banihashemi for pointing this out.

[30]The term "conformal" here refers to the fact that the Hamiltonian $H_\zeta$ generates evolution along the conformal Killing vector $\zeta$, and not to a conformal symmetry of the matter fields (as for CFTs).

[31]We remind the reader that the temperature of a vacuum causal diamond is always $T = -\hbar/2\pi$.

The variations in parenthesis can be combined into a single net variation $\delta\Lambda$, cf. (4.16). Thus, restricting the variations such that $\delta\Lambda = 0$ and holding the volume $V$ fixed, we establish that the stationarity of quantum free energy is equivalent to the stationarity of generalized entropy

$$\delta F_{\text{quan}}\big|_{V,\Lambda} = 0 \qquad \Longleftrightarrow \qquad \delta S_{\text{gen}}\big|_{V,\Lambda} = 0. \tag{4.29}$$

Holding the volume and (effective) cosmological constant fixed is therefore analogous to holding the internal energy fixed in an ordinary thermodynamic system. In fact, it corresponds here to holding fixed the metric and $\Lambda$ contributions to the conformal Hamiltonian $H_\zeta$ (assuming, for non-conformal matter, that there exists a Hamiltonian such that $\delta H_\zeta^\Lambda = (V_\zeta/8\pi G)\,\delta\Lambda$, where $\Lambda$ is the effective cosmological constant (4.16) in a small diamond.[32] In this sense, the entanglement equilibrium hypothesis would correspond to the statement that generalized entropy is stationary at fixed "internal energy" (as it should be for an equilibrium state).

In standard thermodynamical equilibrium, not only is entropy stationary at fixed energy, but also energy is stationary at fixed entropy. Let us now see to what extent the latter is true for causal diamonds. For conformal matter, and for any sized diamond, if the generalized entropy is kept fixed then Eq. (4.26) implies that the Hamiltonian associated to g and $\Lambda$ is stationary in the vacuum

$$\delta F_{\text{quan}}\big|_{S_{\text{gen}}} = 0 \qquad \Longleftrightarrow \qquad \delta H_\zeta^{\text{g}+\Lambda}\big|_{S_{\text{gen}}} = 0. \tag{4.30}$$

For non-conformal matter, we must restrict to small diamonds if we want to hold the generalized entropy fixed, and as above assume the existence of $H_\zeta^\Lambda$. But if instead we keep the Bekenstein-Hawking entropy fixed, rather than the full generalized entropy, then we need not restrict to small diamonds and the *total* Hamiltonian is stationary in the vacuum, i.e.

$$\delta F_{\text{quan}}\big|_{S_{\text{BH}}} = 0 \qquad \Longleftrightarrow \qquad \delta\big(\langle H_\zeta^{\tilde{\text{m}}}\rangle + H_\zeta^{\text{g}+\Lambda}\big)\big|_{S_{\text{BH}}} = 0. \tag{4.31}$$

This equilibrium condition states that the total conformal Killing energy is stationary in the vacuum if the dimension of the Hilbert space is fixed (if we interpret the Bekenstein-Hawking entropy as the logarithm of that dimension). It would be interesting to explore this energy relation further.

Next, we return to the role of the fixed volume constraint in (4.29). For small diamonds the free energy variation at fixed $\Lambda$ can be expressed as

$$\frac{1}{T_H}\delta F_{\text{quan}}\big|_\Lambda = \frac{1}{4G\hbar}(\delta A - k\,\delta V)\big|_\Lambda + \delta S_{\tilde{\text{m}}}, \tag{4.32}$$

(where the volume variation comes from the variation of $H_\zeta^{\text{g}}$ in (4.26)) whereas the variation of the free energy at fixed $V$ and $\Lambda$ is given by

$$\frac{1}{T_H}\delta F_{\text{quan}}\big|_{V,\Lambda} = \frac{1}{4G\hbar}\delta A\big|_{V,\Lambda} + \delta S_{\tilde{\text{m}}}. \tag{4.33}$$

Now any given variation at fixed $\Lambda$ can be expressed as a variation at fixed $V$ and $\Lambda$, together with a variation induced by a diffeomorphism that changes the volume. Under such a diffeo-induced variation, $\delta A - k\delta V$ vanishes, as explained in Sec. 3.3.2, and $\delta S_{\tilde{\text{m}}}$ vanishes because the renormalized matter entropy is zero in the vacuum. The two free energy variations are therefore *equivalent*, i.e.

$$\delta F_{\text{quan}}\big|_\Lambda \sim \delta F_{\text{quan}}\big|_{V,\Lambda}, \tag{4.34}$$

where the equivalence is modulo diffeo-induced variations. This implies that the stationarity of the free energy at fixed $\Lambda$ is equivalent to the stationarity of the free energy at fixed $V$ and

---

[32]It is not clear to us whether this Hamiltonian exists, since $\Lambda$ includes both a local cosmological constant and a piece from the non-conformal matter. The latter contribution seems to spoil the derivation of equation (3.44).

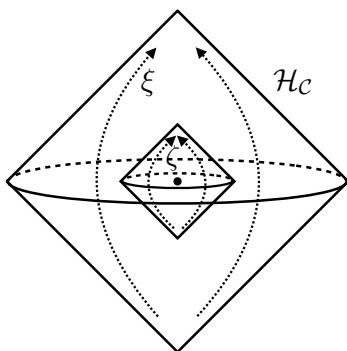

Figure 3: A causal diamond in the de Sitter space static patch. The conformal Killing vector $\zeta$ turns into the timelike Killing vector $\xi$ if the boundary of the diamond coincides with the cosmological horizon $\mathcal{H}_C$.

$\Lambda$, and hence — using the equivalence (4.29) — equivalent to the entanglement equilibrium postulate. Therefore, the derivation of the Einstein equation in [21] can be rephrased without the requirement to fix the volume. One can thus take as the input assumption that the free energy is stationary at fixed $\Lambda$, rather than that the generalized entropy is stationary at fixed $V$ and $\Lambda$ (see also Ref. [76]).

# 5 Limiting cases: small and large causal diamonds

In this section we comment on various limiting cases of the first law of causal diamond mechanics (3.45). Since the law applies to arbitrary sized diamonds in (A)dS, it has a wide domain of applicability. Here we apply it to the static patch of de Sitter spacetime, small diamonds in any maximally symmetric spacetime, flat and AdS Rindler spacetimes, and to the so-called "Wheeler-DeWitt patch" in AdS, tying these limiting cases together into one framework.

## 5.1 De Sitter static patch

If the boundary of a causal diamond in dS space coincides with the cosmological horizon, i.e. if $R = L$, then the conformal Killing vector (2.6) becomes the time translation Killing vector of the static patch,[33]

$$\xi^{\mathrm{dS}} = L\partial_t, \tag{5.1}$$

normalized so that the surface gravity is unity (see Fig. 3). Since this is a true Killing vector, the variation of the gravitational part of the Hamiltonian (3.25) vanishes. This is consistent with (3.35), because the Sitter horizon has extremal area on the $(d-1)$-sphere, so $k = 0$. The first law (3.45) thus reduces to

$$\delta H_\xi^{\tilde{\mathrm{m}}} = -\frac{1}{8\pi G}\left(\kappa\,\delta A + V_\xi\,\delta\Lambda\right). \tag{5.2}$$

The thermodynamic volume (3.14) in this case reduces to $V_\xi^{\mathrm{dS\text{-}static\text{-}patch}} = \kappa L V_L^{\mathrm{flat}}$. The relation (5.2) with $\delta\Lambda = 0$ was established long ago by Gibbons and Hawking [5], and was generalized to include a variation of the cosmological constant in [57, 77].

---

[33]In this section we use the letter $\xi$ for Killing vectors and retain $\zeta$ for conformal Killing vectors.

If one assigns a negative temperature $T = -\kappa\hbar/2\pi$ and pressure $p = -\Lambda/8\pi G$ to the dS static patch, the first law can be turned into a proper thermodynamic relation[34]

$$\delta H_\xi^{\tilde{m}} = T\delta S_{\text{BH}} + V_\xi\,\delta p\,. \tag{5.3}$$

Since this thermodynamic relation is of the form $dH = TdS + V^{\text{th}}dp$, where $H$ is the enthalpy of the system and $V^{\text{th}}$ is the thermodynamic volume, $H_\xi^{\tilde{m}}$ coincides with the enthalpy instead of simply the internal energy. The matter Hamiltonian vanishes in the background, hence we observe that the enthalpy of the static patch of dS space is zero.

Through a Legendre transformation, $U = H - pV^{\text{th}}$, the first law can be rewritten in the standard form $dU = TdS - p\,dV^{\text{th}}$. But what is the internal energy $U$ of de Sitter space? The common lore, cf. e.g. [5], is that the energy of dS space is zero, because it has no asymptotic infinity. However, we find that its (vacuum) internal energy is nonzero and given by $U_{\text{vac}} = \rho\,V_\xi$, the redshifted vacuum energy, where we used the equation of state $p = -\rho$ and the fact that $H = 0$ for dS. The first law can thus be expressed as

$$\delta U = T\delta S_{\text{BH}} - p\,\delta V_\xi\,, \tag{5.4}$$

where $U = U_{\text{vac}} + H_\xi^{\tilde{m}}$ is the internal energy. Finally, we note that the Gibbs free energy of dS space is $G = H - TS = -TS$, and the Helmholtz free energy is $F = U_{\text{vac}} - TS$. As usual, the former is extremized in a fixed $(T, p)$ ensemble, whereas the latter is extremized in a fixed $(T, V_\xi)$ ensemble. The free energy computed in [5] from the on-shell Euclidean action agrees with the Gibbs free energy, and not the Helmholtz free energy, because the Euclidean action was extremized there at fixed period (i.e. fixed temperature) and fixed cosmological constant.[35]

## 5.2 Small diamonds and Minkowski space

In the small radius limit $R \ll L$ the mean curvature (2.12) and the thermodynamic volume (3.14) are given up to second order in $R/L$ by

$$k = \frac{d-2}{R}\left(1 - \frac{1}{2}\frac{R^2}{L^2} + \dots\right), \tag{5.5}$$

$$V_\zeta = \frac{\kappa\,\Omega_{d-2}R^d}{d^2-1}\left(1 + \frac{d}{d+3}\frac{R^2}{L^2} + \dots\right). \tag{5.6}$$

To first order in $R/L$, the first law (3.45) thus reduces to the one that would be found in flat spacetime,

$$\delta H_\zeta^{\tilde{m}} = -\frac{\kappa}{8\pi G}\left[\delta A - \frac{d-2}{R}\,\delta V^{\text{flat}} + \frac{\Omega_{d-2}R^d}{d^2-1}\,\delta\Lambda\right]. \tag{5.7}$$

This identity, without the cosmological constant term, is the one derived in [21], both by Riemann normal coordinate expansion, and by varying the Noether current for the conformal

---

[34]For $\delta H_\xi^{\tilde{m}} = 0$, it might look like one could assign a positive temperature to the dS static patch, i.e. $T = T_{\text{H}} > 0$; however, according to the first law, the "pressure" would then have to be defined as $p = +\Lambda/8\pi G$, in contradiction to the sign of the pressure associated to the cosmological constant when viewed as a stress-energy contribution.

[35]If one takes the timelike Killing vector to be $\xi^{\text{dS}} = \partial_t$, so that $\xi^2 = -1$ at the center of the diamond, then temperature and pressure are not independent in the dS static patch. That is because the surface gravity is set by the dS radius in this case, i.e. $\kappa_{\text{dS}} = 1/L$, and the pressure is determined by the cosmological constant $\Lambda = (d-1)(d-1)/2L^2$. Hence, by fixing the temperature one also fixes the pressure, and vice versa, when using this normalization of the Killing field to define the temperature.

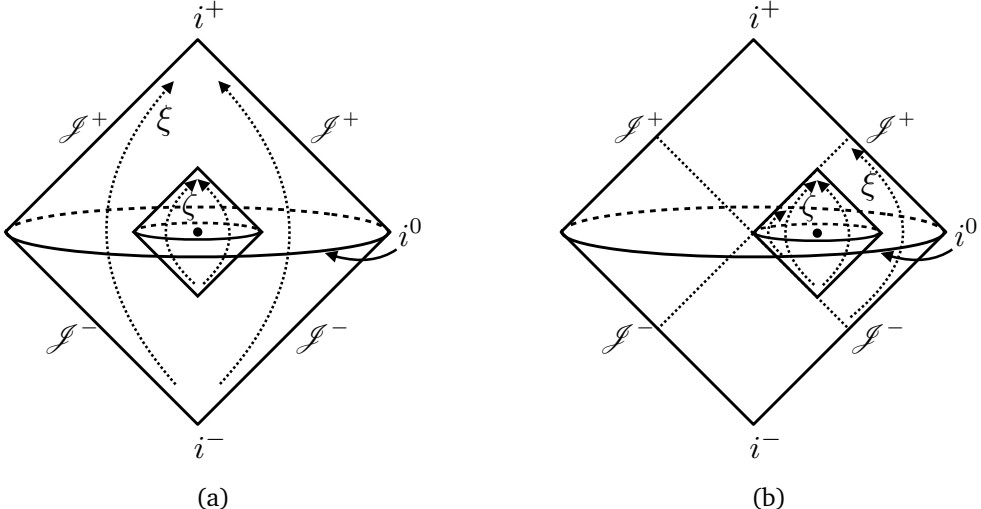

Figure 4: (a) A causal diamond associated to a ball at the center of Minkowski space. If one the normalizes the conformal Killing vector that preserves the diamond such that $\zeta^2 = -1$ at the center of the ball, then it becomes identical to the timelike Killing vector $\xi = \partial_t$ in the infinite-size limit. (b) A causal diamond whose edge touches the bifurcation surface of two Rindler horizons in Minkowski space. In the infinite-size limit the diamond coincides with the right Rindler wedge, and the conformal Killing vector $\zeta$ becomes the boost Killing vector $\xi$.

Killing vector that preserves a causal diamond in flat space. That conformal Killing vector can be recovered from (2.6) in the limit $L \to \infty$, and is given by

$$\zeta^{\text{flat}} = \frac{1}{2R}\left[\left(R^2 - t^2 - r^2\right)\partial_t - 2tr\partial_r\right]. \tag{5.8}$$

(See Fig. 4a for a conformal diagram of a diamond in Minkowski space.) As a side remark, if the variation $\delta T^{\tilde{\text{m}}}_{ab}u^a u^b$ is constant on $\Sigma$,[36] then the conformal Killing energy variation becomes proportional to the thermodynamic volume

$$\delta H^{\tilde{\text{m}}}_\zeta = \delta T^{\tilde{\text{m}}}_{ab}u^a u^b V_\zeta , \tag{5.9}$$

where $\delta T^{\tilde{\text{m}}}_{ab}u^a u^b$ is evaluated at the center of the ball.

## 5.3 Rindler space

A Rindler wedge is an infinite diamond in flat space, as can be seen from the Penrose diagram in Fig. 4b. More precisely, the right Rindler wedge can be obtained by inflating the causal diamond whose edge touches the origin of flat space and whose center is located at $\{t = 0, x^1 = R, x^2 = 0, \ldots, x^{d-1} = 0\}$.[37] In the infinite $R$ limit, only an infinitesimal solid angle of the edge coincides with the entire Rindler horizon, and the rest of the edge covers half the sphere at spatial infinity (see Fig. 5). Moreover, by replacing the coordinate $x^1$ by $x^1 - R$ in $\zeta^{\text{flat}}$ and then taking the limit $R \to \infty$, the conformal Killing vector (5.8) becomes the boost Killing vector of Rindler space,

$$\xi^{\text{Rindler}} = x^1 \partial_t + t \partial_{x^1} . \tag{5.10}$$

---

[36]Note that since $\Sigma$ has vanishing extrinsic curvature, $m^c \nabla_c u^a = 0$ for any vector $m^c$ tangent to $\Sigma$. Thus constancy on $\Sigma$ of the scalar $\delta T^{\tilde{\text{m}}}_{ab}u^a u^b$ is equivalent to the condition $(m^c \nabla_c \delta T^{\tilde{\text{m}}}_{ab})u^a u^b = 0$.

[37]Instead of increasing the size of the diamond to infinity, one could also directly relate the diamond nestled in the corner of the right Rindler wedge to the entire wedge itself through a conformal map. See Appendix E for further details.

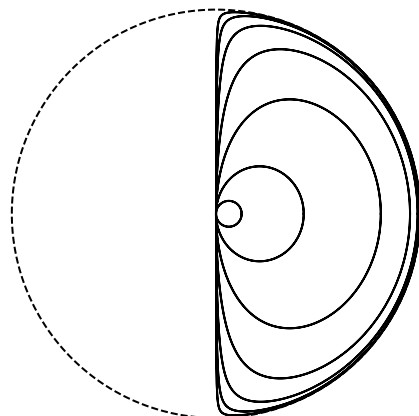

Figure 5: Sequence of causal diamond edges anchored at the origin and with growing radius, shown on a time slice of compactified 2+1 dimensional Minkowski spacetime. In the infinite radius limit, the entire Rindler horizon subtends an infinitesimal angle of the diamond's edge, and the remainder of the edge lies at spatial infinity.

Given that the regions agree it is interesting to understand the relation between the infinite-size limit of the first law of this causal diamond and the first law for the Rindler wedge [10, 11, 78, 79].

Although the first law of causal diamonds was originally derived for diamonds which are centered at the origin of flat space, it actually applies to any causal diamond associated to a spherical region whose center is located at $\{t = t_0, x^i = x_0^i\}$. Hence, in particular it applies to the causal diamond described above which is centered at $x^1 = R$. We evaluated the terms in the first law (5.7) for the variation produced by a point mass $m$ at a distance $\delta$ from the Rindler horizon at $t = 0$, using the weak field approximation and with $\delta\Lambda = 0$. In the infinite $R$ limit the area and volume variations both diverge as $mR$, and these divergences cancel each other. The $k\delta V$ term contains a left-over finite part equal to $m\delta$ which balances the boost Killing energy variation of the Rindler wedge. Strangely, at least when defined by this limiting procedure, the first law for the Rindler wedge involves not only an area variation but also a volume variation, and the finite geometric variation comes entirely from the volume term. Using the fact that the combination $\delta A - k\delta V$ is diffeomorphism invariant — as shown in Section 3.3.2 — this finite contribution could presumably be shifted entirely onto the area term by composing the variation with an appropriate infinitesimal diffeomorphism.

The presence of the volume term in the infinite-size limit is unexpected, since the conformal Killing vector of the diamond becomes the Rindler boost Killing vector (5.10) in this limit, and the gravitational Hamiltonian variation $\delta H_\xi^g$ (3.35) vanishes for a Killing vector $\xi$. Nevertheless, since the volume variation diverges, $k\delta V$ survives (and diverges) in this limit. On the other hand, one can derive an *equilibrium variation* version of the first law for the Rindler wedge by applying the Noether identity (3.26) for the boost Killing vector (5.10) in a finite region, and then taking the infinite region limit. In this way, no volume variation term would be present. For example, for a finite half-spherical ball with radius $R$ and center $o$ on the Rindler horizon the identity takes the form

$$\delta H_\xi^m = -\frac{\kappa}{8\pi G}\delta A_{\text{disk}} + \int_{\text{hemisphere}} (\delta Q_\xi - \xi \cdot \Theta), \tag{5.11}$$

where $\delta A_{\text{disk}}$ is the area variation of the disk on the Rindler horizon and the boundary integral on the right-hand side is over the hemispherical boundary of the half ball. We evaluated the terms in this identity for a weak field variation produced by a point mass at a perpendicular

distance $\delta$ from $o$, in $d = 4$ dimensions. The disk contribution is $-mR/2 + m\delta/2$, while the hemisphere contribution is $mR/2 + m\delta/2$, up to $\mathcal{O}(1/R)$ corrections. In the infinite $R$ limit, the half-spherical ball fills the entire Rindler wedge at $t = 0$, and these two contributions add to $m\delta$, which equals the boost energy variation $\delta H_\xi^{\mathrm{m}}$. This result too is peculiar. We would have expected a first law with just the area variation of the Rindler horizon, as in the physical process version of the first law for Rindler horizons, but that is not the result of this limiting procedure.

A *physical process* version of the first law for Rindler space was stated in [10], and subsequently proven using the Einstein equation in [11,78,79]. This version of the first law describes how the horizon area changes if an infinitesimal amount of Killing energy flows through the horizon. The physical process first law is

$$\delta E_\xi = \frac{\kappa}{8\pi G}\delta A, \tag{5.12}$$

where $\delta A$ is the horizon area change and $\delta E_\xi = \int_{\mathcal{H}} \delta T_{ab}\xi^a d\Sigma^b$ is the flux of Killing energy across the horizon. This differs in two ways from the equilibrium version of the first law (5.11): the hemispherical boundary integral is absent, and the sign of the coefficient of the area variation term is opposite. In the physical process version it is assumed that the energy flux crosses the horizon rather than escapes to infinity, which perhaps accounts for the fact that only the area change of the Rindler horizon enters. As for the sign difference, this is due to a different definition of the area change: the equilibrium version compares the areas of the bifurcation surface in two infinitesimally nearby solutions (Rindler space and the spacetime perturbed by the presence of matter), whereas the process version involves the difference between the asymptotic area of the future horizon and the area of the bifurcation surface in one solution involving a matter boost energy flux across the horizon.

The positive coefficient in the process version (5.12) arises because the generators of the horizon are parallel at future null infinity (due to the teleological nature of the horizon), so that the sign of the area change is the same as the same as the sign of the boost energy flux. For example, when matter with positive Killing energy crosses the future horizon, the horizon expansion decreases because of the attractive nature of gravity. In order to satisfy the zero expansion boundary condition at infinity, the generators must initially have positive expansion, and hence the area of the horizon cross-section *increases* towards the future.[38]

Finally, note that the area change in the process version can also be expressed as a comparison of two solutions,

$$(\delta A)_{\mathrm{phys.\ proc.}} = A_{\mathrm{final}} - A_{\mathrm{initial}} = \delta A_{\mathrm{final}} - \delta A_{\mathrm{initial}}, \tag{5.13}$$

since the area change is zero in the background Minkowski solution. If we consider variations to solutions with the final horizon area held fixed, then $(\delta A)_{\mathrm{phys.\ proc.}} = -\delta A_{\mathrm{initial}}$. With this replacement, the sign of the coefficient in the process version becomes the same as that in the equilibrium version.[39]

## 5.4 AdS-Rindler space

In empty AdS space there also exists a Rindler wedge, which admits a boost Killing vector. Unlike in Minkowski space, accelerating observers in AdS start and end on the boundary at

---

[38]Ref. [79] also interpreted this in terms of an equilibrium variation version of the first law, which refers to the variation of the asymptotic horizon area (assuming the bifurcation surface area is fixed) rather than the variation of the bifurcation surface as in our version.

[39]For another viewpoint on the negative sign, note that the process version for the *past* horizon takes the form: $\delta E_\xi = -\kappa \delta A/8\pi G$, where $\kappa$ is positive and Killing energy flows through the past horizon into the right Rindler wedge. For example, if the Killing energy flux is positive, then the horizon generators converge, so that the area of the past horizon decreases, hence $\delta A = A_{\mathrm{final}} - A_{\mathrm{initial}}$ is negative.

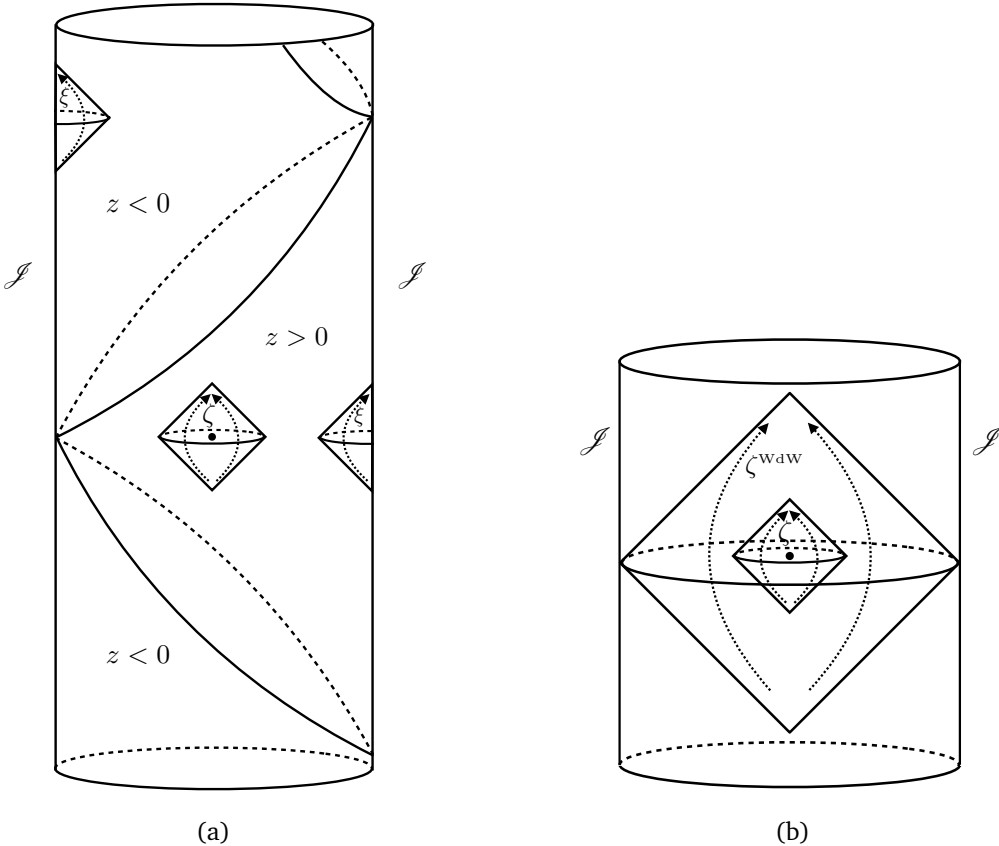

Figure 6: (a) A causal diamond centered at the origin of AdS space can be mapped to two equally large AdS-Rindler wedges in two different Poincaré patches. The situation is depicted for a coordinate radius $R < L/\sqrt{3}$ of the centered ball, since for larger values $R > L/\sqrt{3}$ the diamond overlaps with the right Rindler wedge. (b) A small causal diamond and infinite size diamond in AdS space. The infinite size diamond is called the Wheeler-DeWitt patch of global AdS, and is also preserved by a conformal Killing vector.

a finite global Killing time. In the Penrose diagram the AdS-Rindler wedge therefore has the shape of a half diamond rather than a full diamond (see Fig. 6a). The vertices of the diamond are located at the AdS boundary, and are separated by an infinite proper time but a finite global Killing time. Due to the presence of the conformal boundary, in addition to the area variation one should take the boundary term $\int_\infty [\delta Q_\xi - \xi \cdot \theta(g, \delta g)]$ at spatial infinity into account in the variational identity (3.26). This boundary term has been shown in [80, 81] to be equal to the gravitational energy variation $\delta E_\zeta^g = \int_\infty \delta T_{ab}^g \zeta_{\text{flat}}^a d\Sigma^b$, where $T_{ab}^g$ is the holographic stress-energy tensor, and $\zeta^{\text{flat}}$ is the boundary conformal Killing vector given by (5.8). The first law for AdS-Rindler space can subsequently be derived along the lines of Sec. 3.2 using Wald's variational identity, but now applied to the boost Killing vector.

The variational identity was established in [13] for general relativity, and later generalized in [17] to an arbitrary higher-derivative theory of gravity. The bulk modular energy term $\delta H_\xi^{\tilde{m}}$ was included in the first law in [18, 82], and it has been extended to allow for variations of the cosmological constant in [58, 59]. For general relativity the full first law reads

$$\delta \bar{E}_\zeta^g = \frac{1}{8\pi G}\left(\kappa \, \delta A + \bar{V}_\xi \, \delta \Lambda\right) + \delta H_\xi^{\tilde{m}}. \tag{5.14}$$

The proper volume term is absent since the gravitational Hamiltonian variation $\delta H_\xi^g$ vanishes for a true Killing vector. In the energy variation $\delta \bar{E}_\zeta^g := \int_\infty [\delta Q_\xi - \delta_\Lambda Q_\xi^{\text{AdS}} - \xi \cdot \theta(g, \delta g)]$ we

have subtracted the boundary term in the AdS background due to a variation of $\Lambda$, such that it stays finite when the cosmological constant is varied.[40] From the RHS of the equation we have also subtracted the outer boundary integral of the Noether charge variation $\delta_\Lambda Q_\xi^{\text{AdS}}$, which for general relativity is given by

$$\delta_\Lambda Q_\xi^{\text{AdS}} = -\frac{d-2}{2\Lambda} Q_\xi^{\text{AdS}} \delta\Lambda = \frac{\delta\Lambda}{8\pi G}\, \omega_\xi^{\text{AdS}}, \tag{5.15}$$

where $\omega_\xi^{\text{AdS}}$ is the Killing potential form in AdS. This first equality is due to the scaling $Q_\xi^{\text{AdS}} \sim L^{d-2}$, and the second equality follows from the definition of the Noether charge $j_\xi = dQ_\xi$ and the Killing potential form $\xi \cdot \epsilon = d\omega_\xi$, together with equations (3.2) and (3.11). This amounts to replacing the thermodynamic volume by the background subtracted thermodynamic volume: $\bar{V}_\xi = \int_\infty (\omega_\xi - \omega_\xi^{\text{AdS}}) - \int_{\mathcal{H}} \omega_\xi$ (similar to the expression in footnote 9). Since $\omega_\xi = \omega_\xi^{\text{AdS}}$ for the AdS-Rindler wedge, the boundary term at infinity cancels and only the horizon surface integral remains. More explicitly, in [58] the background subtracted thermodynamic volume was found to be equal to: $\bar{V}_\xi = -\kappa A L^2/(d-1)$, where $A$ is the area of the bifurcation surface of the AdS-Rindler horizon.

As in the Rindler case, although there exists a map from the finite causal diamond to the AdS-Rindler wedge that takes the conformal Killing vector of the former to the true Killing vector of the latter, the relation between the first laws for these two regions is not straightforward. Here we will just describe the nature of the map from a causal diamond in AdS space to an AdS-Rindler wedge. This map consists of shifting the diamond towards the boundary in the $z$ direction, where $z$ is the radial Poincaré coordinate. At the level of the (conformal) isometries, the conformal Killing vector which preserves the diamond transforms under this shift into the boost Killing vector of AdS-Rindler space. In Appendix D we find the Poincaré coordinate expression for the conformal Killing vector that preserves a diamond in AdS with center located at $\{t = 0, z = L, x^i = 0\}$ (see equation (D.17))

$$\zeta^{\text{AdS}} = \frac{1}{2R}\Big[\sqrt{L^2+R^2}(2z\partial_t + 2t\partial_z) - (L^2 + t^2 + \vec{x}^2 + z^2)\partial_t - 2tx^i\partial_i - 2tz\partial_z\Big]. \tag{5.16}$$

Now, by shifting the radial coordinate as $z \to z + \sqrt{L^2 + R^2}$, the conformal Killing vector above turns into the boost Killing vector of an AdS-Rindler wedge [17]

$$\xi^{\text{AdS-Rindler}} = \frac{1}{2R}\Big[\big(R^2 - t^2 - \vec{x}^2 - z^2\big)\partial_t - 2tx^i\partial_i - 2tz\partial_z\Big]. \tag{5.17}$$

At the boundary ($z = 0$) this reduces to the conformal Killing vector $\zeta^{\text{flat}}$ in (5.8). The boost Killing vector becomes null on the Killing horizon $\mathcal{H} = \{z^2 = (R \pm t)^2 - \vec{x}^2\}$, and vanishes at the vertices $\{t = \pm R, z = x^i = 0\}$ and hemisphere $\mathcal{B} = \{t = 0, z^2 + \vec{x}^2 = R^2\}$. Notice, though, that there are two solutions to these quadratic equations, one for which $z$ takes positive values and the other for which $z$ takes negative values. Thus, one part of the causal diamond is mapped to the AdS-Rindler wedge in the $z > 0$ Poincaré patch, and the other part is shifted to the equally large Rindler wedge in the $z < 0$ Poincaré patch (see Fig. 6a).

## 5.5 Wheeler-DeWitt patch in AdS

The first law applies also to causal diamonds whose spatial slice is an entire slice of AdS. The spacetime region covered by such an infinite diamond is commonly known as the "Wheeler-DeWitt patch" of AdS (see Fig. 6b). In this limit, i.e. $R/L \to \infty$, the vector field $\zeta^{\text{AdS}}$ (5.16)

---

[40] We note that $\theta(g, \delta_\Lambda g) = 0$ since the symplectic potential for general relativity is linear in $\nabla_c \delta g_{ab}$, and we have $\delta_\Lambda g_{ab} = -(\delta\Lambda/\Lambda)g_{ab}$ (see also appendix C of [59]).

takes the following simple form in Poincaré coordinates

$$\zeta^{\text{WdW}} = z\partial_t + t\partial_z. \tag{5.18}$$

This limiting value also follows directly from expression (D.17) for $\zeta^{\text{AdS}}$ in terms of the generators of the conformal group. By taking the limit $R/L \to \infty$ of (3.45), we obtain the first law for the Wheeler-DeWitt patch

$$\delta H_\zeta^{\tilde{\text{m}}} = -\frac{\kappa}{8\pi G}\left[\delta A - \frac{d-2}{L}\delta V\right]. \tag{5.19}$$

In this limit the proper volume $V \approx AL/(d-2)$ of the $t = 0$ timeslice in AdS is divergent, but its variation can be finite. Here we have restricted to fixed cosmological constant, i.e. $\delta\Lambda = 0$. Note that the proper volume variation is still present in the first law for the Wheeler-dDeWitt patch, because the conformal Killing vector (5.18) is not a true Killing vector.

# 6 Discussion

In this paper we explored aspects of the gravitational thermodynamics of causal diamonds in maximally symmetric spacetimes and their first order variations. Our starting point was the notion that the maximally symmetric diamonds behave as thermodynamic equilibrium states. This is initially motivated by the examples offered by the static patch of de Sitter spacetime and the Rindler wedge of Minkowski spacetime, which are special cases of causal diamonds admitting a true Killing field. Other maximally symmetric causal diamonds, in particular finite ones, admit only a conformal Killing vector. Since neither general relativity nor ordinary matter are conformally invariant, it is not at all clear from the outset that the presence of a conformal Killing symmetry should be adequate to support the interpretation of a physical equilibrium state. However, it seems in all respects to be sufficient. This can be traced to three important facts: (i) the conformal Killing vector is an "instantaneous" Killing vector at the maximal volume slice, which (ii) behaves at the edge as a boost-like Killing vector, with a well-defined surface gravity; and (iii) in a maximally symmetric diamond the vacuum of a conformal matter field restricted to the diamond is thermal with respect to the Hamiltonian that generates the conformal Killing flow.

We first established a classical Smarr formula and a "first law" variational identity for causal diamonds in maximally symmetric spacetime, i.e. in either Minkowski, de Sitter, or Anti-de Sitter spacetime. Since we include a cosmological constant and variations thereof, the "thermodynamic volume" [33–35] plays a role, generalized here to the case of a conformal Killing vector. The name is appropriate for this quantity since it is thermodynamically conjugate to the cosmological constant which is a type of pressure. It is defined, given a (conformal) Killing vector $\zeta$, by $V_\zeta = \int_\Sigma \zeta \cdot \epsilon$, which might well be called the "redshifted volume". For finite causal diamonds it appears as such, while for infinite asymptotically Anti-de Sitter diamonds or black hole spacetimes it diverges. We reviewed how finite relations have nevertheless been obtained by subtracting the same divergence from both sides of an equation.

We then analyzed the thermodynamic interpretation of the first law, finding that the gravitational temperature of a diamond is *minus* the Hawking temperature associated with the horizon of the conformal Killing vector. The idea that a negative temperature should be assigned to the static patch of de Sitter spacetime has been floated before [31], but not followed up. We found that the notion appears sound, and indeed is required by the thermodynamics of causal diamonds in general. The consistency with the *positive* Gibbons-Hawking temperature hinges on the fact that, in gravitational thermodynamics, the matter contribution to the first law

enters *either* as an energy contribution, *or* as an entropy contribution, unlike in ordinary thermodynamics. In the latter form, the matter entropy combines with the Bekenstein-Hawking area entropy to comprise the generalized entropy. We showed that it works this way also for quantum corrections. In establishing this for nonconformal matter we invoked a conjecture, previously postulated in [21] and checked in [37,38], to relate the conformal boost energy to the entanglement entropy.

We next reformulated the entanglement equilibrium proposal of [21], replacing the role of generalized entropy maximization by a free energy extremization, and showed that in this way, the need to fix the volume is no longer present. We find that extremization of the free conformal energy $H_\zeta - T S_{\text{BH}}$ at fixed cosmological constant is equivalent to the stationarity of the generalized entropy at fixed cosmological constant and fixed volume. Therefore, the Einstein equation is implied by stationarity of the free energy at fixed cosmological constant, without fixing the volume.

In the final section we considered limiting cases of causal diamonds, such as small diamonds, the de Sitter static patch, Rindler space, and AdS-Rindler space, and showed how our general result for the first law appears and reduces to known results in these different settings. In particular, we gained some interesting perspective on the first law in Rindler space, viewed as the infinite size limit of a causal diamond.

In one of the appendices we established a link between conformal Killing time and York time in a maximally symmetric causal diamond. The latter time is defined by the mean curvature on a foliation by constant mean curvature slices. We found that these slices coincide with the foliation orthogonal to the conformal Killing vector, and that the conformal Killing time parameter also labels these surfaces.

We end with some questions and future research directions:

(i) Can the first law for causal diamonds be generalized to non-spherical regions and/or non-maximally symmetric spacetimes? The particular form (3.45) applies only to maximally symmetric spaces, but the general form (3.28) holds in general. On the other hand, the conformal Killing Hamiltonian variation may have no particular geometric significance. Since the connection between the York time Hamiltonian and proper volume holds for any solution to Einstein gravity, however, perhaps this would produce a geometrically meaningful form of the first law in the absence of a conformal Killing vector. Note also that the diamond itself doesn't play an essential role in the first law, since all the quantities are evaluated at the ball. Therefore, the thermodynamics might be a property of the ball rather than the full diamond.

(ii) In the case of a large diamond in AdS, could there be a dual CFT interpretation of the first law? Proposals exist for holographic CFT duals of the maximal volume and boundary area in the bulk. The area of the cut-off boundary surface in Planck units may be dual to the number of degrees of freedom of the cut-off CFT [83]. The proposed dual quantities for the maximal volume are computational complexity [84], fidelity susceptibility [85], and (for variations) the symplectic form on the space of sources in the Euclidean path integral [86, 87]. One can also consider varying the cosmological constant, which is equivalent to varying the number of degrees of freedom $N$ in the CFT. The thermodynamic volume is the conjugate to the cosmological constant, and in the AdS-Rindler case it is dual to the chemical potential for $N$ in a ball in the CFT [58]. It would be interesting to find a variational relation in the CFT between these quantities dual to the bulk area, maximal volume, and thermodynamic volume of the maximal slice.

(iii) Black hole thermodynamics, as well as de Sitter thermodynamics, has been well studied from the viewpoint of the partition function, formulated as a path integral over Riemannian geometries beginning with the work of Gibbons and Hawking [5]. Can the

thermodynamics of causal diamonds similarly be formulated in terms of the Euclidean path integral in the canonical and/or microcanonical ensemble? If so this should provide a foundation for determining the stability of different ensembles, and for a systematic treatment of the quantum corrections.

## Acknowledgements

It is a pleasure to thank Batoul Banihashemi, Pablo Bueno, Juan Maldacena, Vincent Min, Maulik Parikh, Antony Speranza, and Erik Verlinde for insightful comments and discussions about this work. TJ was supported in part by NSF grants PHY-1407744 and PHY-1708139. MV acknowledges support from the Spinoza Grant of the Dutch Science Organisation (NWO), and from the Delta ITP consortium, a special program of the NWO. Finally, we thank the organizers of the workshop "Holography for black holes and cosmology" in Brussels and of the "Amsterdam String Workshop", both held in 2016, where this work was initiated.

## A Conformal isometry of causal diamonds in (A)dS

We seek the conformal isometry that preserves a causal diamond in de Sitter space. Since the property of being a conformal isometry is invariant under a Weyl rescaling of the metric, we will use the form of the line element (2.4), leaving off the conformal factor $\text{sech}^2(r_*/L)$. Note that any vector field of the form

$$\zeta = A(u)\partial_u + B(v)\partial_v \tag{A.1}$$

is then a conformal isometry of the $du\,dv$ factor of the metric: $\mathcal{L}_\zeta du\,dv = [A'(u)+B'(v)]du\,dv$, where $\mathcal{L}_\zeta$ is the Lie derivative along $\zeta$. It will be a conformal isometry of the full metric provided $\mathcal{L}_\zeta \sinh^2(r_*/L) = [A'(u) + B'(v)]\sinh^2(r_*/L)$. Using $r_* = (v-u)/2$ we find that in fact $\mathcal{L}_\zeta \sinh^2(r_*/L) = \frac{1}{L}(B-A)\sinh(r_*/L)\cosh(r_*/L)$, so $\zeta$ is a conformal Killing field if

$$[A'(u) + B'(v)]L\tanh(r_*/L) = B(v) - A(u). \tag{A.2}$$

For $u = v$, $r_* = 0$ and thus $\tanh(r_*/L) = 0$, so this implies $B(v) = A(v)$. Then evaluating at $v = 0$, we have $r_* = -u/2$, so (A.2) becomes

$$[A'(u) + A'(0)]L\tanh(u/(2L)) = A(u) - A(0). \tag{A.3}$$

The general solution to this equation is

$$A(u) = B(u) = a + b\sinh(u/L) + c\cosh(u/L). \tag{A.4}$$

To map the diamond onto itself, the flow of $\zeta$ must leave invariant the boundaries $u = -R_*$ and $v = R_*$. This implies $A(\pm R_*) = 0$, and hence $A(u) = a_1(\cosh(u/L) - \cosh(R_*/L))$. The requirement that the surface gravity $\kappa$ of $\zeta$ be unity at the future conformal Killing horizon implies $\kappa = -B'(R_*) = -\frac{1}{L}a_1\sinh(R_*/L) = 1$. Therefore, the conformal Killing vector that preserves a diamond in dS and has unit surface gravity at the horizon is given by

$$\zeta = \frac{L}{\sinh(R_*/L)}\Big[ (\cosh(R_*/L) - \cosh(u/L))\,\partial_u + (\cosh(R_*/L) - \cosh(v/L))\,\partial_v \Big]. \tag{A.5}$$

Expressed in terms of the standard $t$ and $r$ coordinates, $\zeta$ reads

$$\zeta = \frac{L^2}{R}\left[ \Big(1 - \frac{\sqrt{1-(R/L)^2}}{\sqrt{1-(r/L)^2}}\cosh(t/L)\Big)\partial_t - \frac{r}{L}\sqrt{(1-(R/L)^2)(1-(r/L)^2)}\sinh(t/L)\partial_r \right]. \tag{A.6}$$

A similar expression exists for the conformal Killing vector which preserves a causal diamond in AdS, which can be obtained by sending $L \to iL$. Moreover, in the flat space limit $L \to \infty$, $\zeta$ reduces to the well-known expressions [17, 21]

$$\begin{aligned}
\zeta^{\text{flat}} &= \frac{1}{2R} \left[ \left( R^2 - u^2 \right) \partial_u + \left( R^2 - v^2 \right) \partial_v \right] \\
&= \frac{1}{2R} \left[ \left( R^2 - t^2 - r^2 \right) \partial_t - 2rt \partial_r \right].
\end{aligned} \tag{A.7}$$

## B  Conformal Killing time and mean curvature

In this appendix we show that on slices of constant conformal Killing time $s$ in a maximally symmetric causal diamond, the trace $K$ of the extrinsic curvature is constant, and we establish the relation (3.50) between $K$ and $s$. To this end, we first construct a coordinate system adapted to the flow of $\zeta$, and then we compute $K$ on the constant $s$ slices using this coordinate system.

Conformal Killing time is a function $s$ satisfying $\zeta \cdot ds = 1$, but there are many such functions. Given one choice of a constant $s$ hypersurface, $s$ is determined by integration along the flow lines of $\zeta$. We choose the $t = 0$ slice of the diamond to be the $s = 0$ slice. This slice is everywhere orthogonal to $\zeta$, hence $\zeta_a$ and $\nabla_a s$ are parallel on it. The Lie derivative $\mathcal{L}_\zeta \nabla_a s$ vanishes by definition of $s$, and $\mathcal{L}_\zeta \zeta_a = 2\alpha \zeta_a$, since $\zeta$ is a conformal Killing vector satisfying $\mathcal{L}_\zeta g_{ab} = 2\alpha g_{ab}$. The flow therefore preserves the proportionality of these two covectors, hence all of the constant $s$ slices are orthogonal to $\zeta$.

Now choose a spherically symmetric coordinate $x$ on the $s = 0$ slice, such that $|dx| = |ds|$ and $x = 0$ at $r = 0$, and extend it to the diamond by the flow of $\zeta$, i.e. so that $\zeta \cdot dx = 0$. Both $ds$ and $dx$ are invariant under the conformal Killing flow, and they are orthogonal and have equal norms at $s = 0$, hence these conditions hold for all values of $s$. The line element (2.4) therefore takes the form

$$ds^2 = C^2(s, x)(-ds^2 + dx^2) + r^2 d\Omega_{d-2}^2. \tag{B.1}$$

The conformal Killing equation then implies that $r = C\rho$, where $\rho = \rho(x)$ is a function of $x$ alone, so we have

$$ds^2 = C^2(s, x)[-ds^2 + dx^2 + \rho^2(x) d\Omega_{d-2}^2]. \tag{B.2}$$

In this coordinate system the future horizon of the diamond is located at $s = \infty$, and the past horizon is at $s = -\infty$ (see Fig. 2 for an illustration of constant $s$ and $x$ surfaces). Further, we have $\zeta = \partial_s$, the surface gravity is $\kappa = -C^{-1} \partial_s C \big|_{s \to \infty}$, and the unit timelike vector normal to the constant $s$ slices is $u = C^{-1} \partial_s$. The divergence $\nabla_a u^a$ is the trace $K$ of the extrinsic curvature of these slices,

$$K = (d-1)C^{-2} \partial_s C = (1-d) \partial_s C^{-1}. \tag{B.3}$$

It remains to show that $\partial_s C^{-1}$ is independent of $x$, and to evaluate it explicitly.[41]

We proceed by finding the relation between the null coordinates $(u, v)$ defined in (2.2) and the null coordinates $\bar{u} = s - x$ and $\bar{v} = s + x$. The form of the metric indicates that $\bar{u} = \bar{u}(u)$ and $\bar{v} = \bar{v}(v)$, and the conditions $\zeta \cdot ds = 1$ and $\zeta \cdot dx = 0$ imply $\zeta \cdot d\bar{u} = 1 = \zeta \cdot d\bar{v}$. Using the expression (A.5) for $\zeta$ (which is the conformal Killing vector which has unit surface gravity), we find the coordinate relations

$$e^{\bar{u}} = \frac{\sinh[(R_* + u)/2L]}{\sinh[(R_* - u)/2L]}, \quad e^{u/L} = \frac{\cosh[(R_*/L + \bar{u})/2]}{\cosh[(R_*/L - \bar{u})/2]}, \quad \text{and} \quad (u \to v). \tag{B.4}$$

---

[41]We note that the mean curvature can also be expressed as $K = (d-1)\alpha/|\zeta|$, since $\alpha = \nabla \cdot \zeta/d = C^{-1} \partial_s C$ and $|\zeta| = C$.

With the help of Mathematica we then find

$$C^{-1} = \frac{\cosh s + \cosh(x)\cosh(R_*/L)}{L\sinh(R_*/L)}, \qquad \rho = \sinh x. \tag{B.5}$$

With this equation for $\rho$ the metric between brackets in (B.2) becomes that of conformal Killing time cross hyperbolic space, $\mathbb{R} \times \mathbb{H}^{d-1}$.[42] From this expression for $C^{-1}$ it follows that

$$K = \frac{1-d}{L\sinh(R_*/L)}\sinh s = (d-1)\dot\alpha|_{s=0}\sinh s, \tag{B.6}$$

where we used (2.11) in the last equality. This establishes in particular that $K$ is constant on the constant $s$ slices.[43] Instead of using (2.11) we could use $\alpha = \nabla \cdot \zeta/d = -C\partial_s C^{-1}$ to write $\dot\alpha = u^a\nabla_a\alpha = -C^{-1}\partial_s(C\partial_s C^{-1})$. Inserting (B.5) then yields $\dot\alpha|_{s=0} = -1/[L\sinh(R_*/L)]$, which again establishes the fact that $\dot\alpha|_{s=0}$ is independent of $x$.

Finally, as a side comment, let us explain how the conformal Killing flow is related to the "new York" transformation of [86, 87]. In terms of the induced metric and the extrinsic curvature of a Cauchy surface the latter transformations reads: $\delta_Y h_{ab} = 0$ and $\delta_Y K_{ab} = \tilde\alpha h_{ab}$. This transformation is not a diffeomorphism in general, so it is not expected to be the same as the variation induced by the conformal Killing vector: $\delta_\zeta\phi = \mathcal{L}_\zeta\phi$. In our setup the extrinsic curvature is given by $K_{ab} := 2\nabla_{(a}u_{b)} = (C^{-2}\partial_s C)h_{ab} = \alpha h_{ab}/|\zeta|$. The Lie derivative of the induced metric $h_{ab} := u_a u_b + g_{ab}$ on $\Sigma$ and of the extrinsic curvature are given by

$$\mathcal{L}_\zeta h_{ab} = 2\alpha h_{ab}, \qquad \mathcal{L}_\zeta K_{ab} = (C^{-2}\partial_s^2 C)h_{ab} = (\dot\alpha + \alpha^2/|\zeta|)h_{ab}. \tag{B.7}$$

The first equation follows from the definition (2.7) of a conformal Killing vector, and from $\mathcal{L}_\zeta u_a = \alpha u_a$, where $u_a = \zeta_a/|\zeta|$. The second equation follows from the expression above for $K_{ab}$ and from the fact that $h_{ab}(s,x) = C^2(s,x)\sigma_{ab}(x)$, where $\sigma_{ab}(x)$ is the metric on hyperbolic space $\mathbb{H}^{d-1}$. Since $\alpha = 0$ on $\Sigma$, we find that on the extremal slice of the diamond

$$\mathcal{L}_\zeta h_{ab}\big|_\Sigma = 0, \qquad \mathcal{L}_\zeta K_{ab}\big|_\Sigma = \dot\alpha|_{s=0}h_{ab}. \tag{B.8}$$

This is of the same form as the new York transformation, if we identify $\tilde\alpha = \dot\alpha\big|_{s=0}$. In other words, the new York transformation and the conformal Killing transformation only coincide *at* the extremal surface $\Sigma$.

# C  Zeroth law for bifurcate conformal Killing horizons

A vector field $\zeta^a$ is a conformal Killing vector of the metric $g_{ab}$ if $\mathcal{L}_\zeta g_{ab} = 2\alpha g_{ab}$ for some function $\alpha$. A conformal Killing horizon $\mathcal{H}$ is a null hypersurface whose null generators are orbits of a conformal Killing field. The notion of surface gravity for Killing horizons can be extended to conformal Killing horizons [27–29], however definitions that are equivalent in the former case (see e.g. Sec. 12.5 in [88]) are not generally equivalent in the latter case. One of these definitions has two properties not shared by the other definitions: it is (i) Weyl invariant, and (ii) constant on the horizon. In this appendix we establish these properties, the second one assuming $\mathcal{H}$ has a bifurcation surface $\mathcal{B}$ where $\zeta^a$ vanishes. We also show that

---

[42]The limit $L \to \infty$, $R_* \to R$ of (B.4) and (B.5) yields the result for a diamond in flat spacetime: $\bar u = \ln[(R+u)/(R-u)]$, $u = R\tanh(\bar u/2)$, and $C = R/(\cosh s + \cosh x)$. The limit $R \to L$, $R_* \to \infty$ leads to the result for the static patch of de Sitter spacetime: $\bar u = u/L$, and $C = L/\cosh x$. And by replacing $L \to iL$ and setting $R_* = L\pi/2$ we obtain for the Wheeler-DeWitt patch of AdS: $C = L/\cosh s$, where $L$ is the AdS radius.

[43]A degenerate case is the dS static patch ($R_* \to \infty$), since in that case $K = 0$ for all $s$, i.e. all the CMC slices have zero mean curvature, so there is no York time flow for the dS static patch.

$\nabla_a \zeta_b \overset{\mathcal{B}}{=} \kappa n_{ab}$, where $\kappa$ is the surface gravity and $n_{ab}$ is the binormal to $\mathcal{B}$. In particular, the Killing equation holds at $\mathcal{B}$.

The conformal Killing horizon is a hypersurface defined by the equation $\zeta^2 = 0$. The gradient $\nabla_a \zeta^2$ is normal to this surface; and, since it is a null surface, the normal is proportional to $\zeta_a$, so

$$\nabla_a\left(\zeta^b \zeta_b\right) \overset{\mathcal{H}}{=} -2\kappa \zeta_a \tag{C.1}$$

for some function $\kappa$. Under a Weyl transformation, $g_{ab} \rightarrow \Omega^2 g_{ab}$, $\zeta^a$ remains a conformal Killing field, and $\mathcal{H}$ a conformal Killing horizon. Moreover both sides of (C.1) transform homogeneously, acquiring a factor $\Omega^2$. This definition of the surface gravity $\kappa$ is therefore Weyl invariant [29].

The zeroth law for stationary black holes asserts that the surface gravity is constant over the entire event horizon. This was originally proven in [2] by assuming the Einstein field equations and the dominant energy condition for matter.[44] A simpler, and in a sense more general, proof of the zeroth law for black hole horizons was given by Kay and Wald in [91].[45] This proof applies to Killing horizons that contain a bifurcation surface where the Killing vector vanishes — also called *bifurcate Killing horizons* — and it holds independently of any gravitational field equation or energy condition. Here we extend this proof to bifurcate conformal Killing horizons:

**Zeroth Law.** *Let $\mathcal{H}$ be a (connected) conformal Killing horizon with bifurcation surface $\mathcal{B}$. Then the surface gravity $\kappa$, defined in (C.1), is constant on $\mathcal{H}$.*

*Proof:* We show first that $\kappa$ is constant along the generators of $\mathcal{H}$, and then that its value also does not vary from generator to generator.

That $\kappa$ is constant on each generator can be expected, since the flow of the conformal Killing vector leaves the metric unchanged up to a Weyl transformation, and $\kappa$ is Weyl invariant. To make this into a computational proof, we take the Lie derivative of both sides of (C.1) along $\zeta$. On the lhs we have

$$\mathcal{L}_\zeta \nabla_a \zeta^2 = \nabla_a \mathcal{L}_\zeta \zeta^2 = \nabla_a \mathcal{L}_\zeta (g_{bc} \zeta^b \zeta^c) = \nabla_a (2\alpha \zeta^2) \overset{\mathcal{H}}{=} 2\alpha \nabla_a \zeta^2 \overset{\mathcal{H}}{=} -4\alpha \kappa \zeta_a. \tag{C.2}$$

On the rhs we have

$$\mathcal{L}_\zeta(-2\kappa \zeta_a) = (-2\mathcal{L}_\zeta \kappa - 4\alpha\kappa)\zeta_a. \tag{C.3}$$

Since (C.2) and (C.3) must be equal, we conclude that

$$\mathcal{L}_\zeta \kappa \overset{\mathcal{H}}{=} 0, \tag{C.4}$$

i.e. $\kappa$ is constant along the flow of $\zeta^a$ and hence along each null generator of $\mathcal{H}$.

Next, to prove that $\kappa$ does not vary from generator to generator, we will show that it is constant on the bifurcation surface $\mathcal{B}$. We begin by noting that, since $\zeta^a$ is a conformal Killing vector, we have

$$\nabla_a \zeta_b = \alpha g_{ab} + \omega_{ab}, \tag{C.5}$$

where $\omega_{ab} = -\omega_{ba}$ is an antisymmetric tensor. If $m^a$ is tangent to $\mathcal{B}$, then the contraction of this equation with $m^a m^b$ yields zero on the lhs (since $\zeta_b = 0$ on $\mathcal{B}$), and yields $\alpha m^2$ on the rhs. It follows that the conformal factor $\alpha$ vanishes on $\mathcal{B}$, and thus the Killing equation

---

[44]There exists another proof of the zeroth law by Carter [89] and Rácz and Wald [90] that does not depend on the Einstein equation or an energy condition. The proof assumes the black hole horizon is a Killing horizon and the black hole is either (i) static or (ii) stationary-axisymmetric with "$t - \phi$" reflection isometry. It would be interesting to see whether this proof can be generalized to conformal Killing horizons.

[45]Gibbons and Geroch are acknowledged, respectively, in [91] for this version of the zeroth law and for the proof.

$\nabla_{(a}\zeta_{b)} \stackrel{\mathcal{B}}{=} 0$ holds there. In addition, the contraction of the lhs of (C.5) with a single vector $m^a$ vanishes on $\mathcal{B}$, so $m^a \omega_{ab} \stackrel{\mathcal{B}}{=} 0$ for all $m^a$ tangent to $\mathcal{B}$, which implies that $\omega_{ab}$ is proportional to the binormal $n_{ab}$ to $\mathcal{B}$, i.e. $\nabla_a \zeta_b \stackrel{\mathcal{B}}{=} \beta n_{ab}$ for some function $\beta$. To evaluate $\beta$, contract both sides of this equation with a null vector $k^b$ that is tangent to $\mathcal{H}$. On the lhs we have $k^b \nabla_a \zeta_b$, which according to (C.1) off of $\mathcal{B}$ is equal to $-\kappa k_a$ (since $k_a$ is proportional to $\zeta_a$). Taking the limit as $\mathcal{B}$ is approached along the horizon thus yields $k^b \nabla_a \zeta_b \stackrel{\mathcal{B}}{=} -\kappa k_a$. On the rhs we have $\beta n_{ab} k^b = -\beta k_a$, so it follows that $\beta = \kappa$, if the sign of the binormal is chosen so that $n_{ab} k^b = -k_a$.[46] This establishes the useful relation[47]

$$\nabla_a \zeta_b \stackrel{\mathcal{B}}{=} \kappa\, n_{ab}. \tag{C.6}$$

To demonstrate that $\kappa$ is constant on $\mathcal{B}$, we act on both sides of (C.6) with $n^{ab} m^c \nabla_c$. Since $n^{ab} n_{ab} = -2$ is constant on $\mathcal{B}$, we obtain

$$n^{ab} m^c \nabla_c \nabla_a \zeta_b \stackrel{\mathcal{B}}{=} -2m^a \nabla_a \kappa. \tag{C.7}$$

Finally, a conformal Killing vector satisfies a generalization of the usual Killing identity,

$$\nabla_c \nabla_a \zeta_b = \zeta^d R_{dcab} + g_{ab} \nabla_c \alpha + g_{bc} \nabla_a \alpha - g_{ac} \nabla_b \alpha. \tag{C.8}$$

The contraction of the rhs of this identity with $n^{ab} m^c$ vanishes on $\mathcal{B}$, since $\zeta^d$ and $\alpha$ vanish on $\mathcal{B}$, and $n^{ab} m_b = 0$. It follows that the lhs of (C.7) vanishes, which establishes that $\kappa$ is constant on $\mathcal{B}$. $\square$

# D  Conformal group from two-time embedding formalism

It is well-known that $d$-dimensional Minkowski spacetime can be embedded in $\mathbb{R}^{2,d}$ as a section of the light cone through the origin, described by the equation (see e.g. [92])

$$X \cdot X = -\left(X^{-1}\right)^2 - \left(X^0\right)^2 + \left(X^1\right)^2 + \cdots + \left(X^d\right)^2 = 0. \tag{D.1}$$

Here $X^A$ are the standard flat coordinates on $\mathbb{R}^{2,d}$. The embedding space naturally induces a metric on a hyper-lightcone section, for example the section $X^{-1} + X^d = 1$ realizes the standard Minkowski metric on $\mathbb{R}^{1,d-1}$. In fact, any conformally flat manifold can be embedded in $\mathbb{R}^{2,d}$ as a section of the light cone, because under the coordinate transformation $\widetilde{X}^A = \Omega(x) X^A$ the induced metric becomes

$$d\widetilde{s}^2 = (X d\Omega + \Omega dX)^2 = \Omega^2 dX \cdot dX = \Omega^2(x) ds^2, \tag{D.2}$$

where we used the light cone properties $X \cdot dX = 0$ and $X \cdot X = 0$. This means that the induced metrics on two different hyper-lightcone sections are related by a Weyl transformation.

This embedding construction is particularly useful for characterizing the conformal group $O(2, d)$ of Minkowski space, since it corresponds to the symmetry group that preserves a light cone in $\mathbb{R}^{2,d}$. Moreover, it follows from the observation above that the conformal group generators of *any* conformally flat spacetime can be obtained from the embedding space. In this appendix we will use the two-time embedding formalism to derive all conformal Killing vectors of dS and AdS space – which are both conformally flat – and, in particular, we will write the conformal Killing vector (A.6) in terms of embedding coordinates.

---

[46]Note that this means that the sign of $\kappa$ is opposite for the two sheets of a bifurcate Killing horizon.

[47]This relation is also needed to derive the expressions (3.6) and (3.27) for the Noether charge (see section 7 in [43]).

### D.1 Conformal Killing vectors of de Sitter space

The embedding coordinates for the static line element (2.1) of de Sitter space are

$$
\begin{aligned}
X^0 &= \sqrt{L^2 - r^2}\,\sinh(t/L), & X^{-1} &= L, \\
X^d &= \sqrt{L^2 - r^2}\,\cosh(t/L), & X^i &= r\,\Omega^i, \quad i = 1,\dots,d-1,
\end{aligned}
\tag{D.3}
$$

with $\Omega^1 = \cos\theta_1, \Omega^2 = \sin\theta_1\cos\theta_2, \dots, \Omega^{d-1} = \sin\theta_1\sin\theta_2\cdots\sin\theta_{d-2}$ and the condition $\sum_{i=1}^{d-1}(\Omega^i)^2 = 1$. Note that we have promoted the de Sitter length scale to a coordinate, which is a convenient trick to obtain the conformal group. If the relation $X^{-1} = L$ is inserted back into (D.1), then the embedding constraint turns into the equation for a hyperboloid in $\mathbb{R}^{1,d}$, which is the standard embedding of de Sitter space.

The conformal group of $d$-dimensional Lorentzian de Sitter space is $O(2,d)$, whereas the isometry group is $O(1,d)$. Hence, de Sitter space admits $\frac{1}{2}d(d+1)$ true Killing vectors and $d+1$ conformal Killing vectors which do not generate isometries (in total there are $\frac{1}{2}(d+1)(d+2)$ conformal generators). The generators of $O(2,d)$ are boosts and rotations in embedding space, and hence take the form

$$
J_{AB} = i\left(X_A\partial_{X^B} - X_B\partial_{X^A}\right).
\tag{D.4}
$$

Coordinates with lower indices are defined as $X_A = \eta_{AB}X^B$, such that for instance the generator $J_{01} = -i\left(X^0\partial_{X^1} + X^1\partial_{X^0}\right)$ is a proper boost. The generators satisfy the usual Lorentz algebra commutation relations

$$
[J_{AB}, J_{CD}] = i\left(\eta_{AD}J_{BC} + \eta_{BC}J_{AD} - \eta_{AC}J_{BD} - \eta_{BD}J_{AC}\right).
\tag{D.5}
$$

In embedding space the true Killing vectors correspond to boosts and rotations in the $X^0,\dots,X^d$ directions, since these generators preserve the light cone section, whereas the extra $d+1$ conformal Killing vectors are boosts and rotations in the $X^{-1}$ direction, which do not preserve the section.

We are now ready to compute the conformal generators explicitly. In terms of static coordinates the true Killing vectors of de Sitter space are given by

$$
\begin{aligned}
iJ_{0d} &= L\partial_t \\
iJ_{0i} &= \frac{Lr\Omega_i\cosh(t/L)}{\sqrt{L^2-r^2}}\partial_t + \sqrt{L^2-r^2}\,\Omega_i\sinh(t/L)\partial_r + \frac{\sqrt{L^2-r^2}}{r}\sinh(t/L)\nabla_i \\
iJ_{id} &= \frac{Lr\Omega_i\sinh(t/L)}{\sqrt{L^2-r^2}}\partial_t + \sqrt{L^2-r^2}\,\Omega_i\cosh(t/L)\partial_r + \frac{\sqrt{L^2-r^2}}{r}\cosh(t/L)\nabla_i \\
iJ_{ij} &= \Omega_j\partial_{\Omega^i} - \Omega_i\partial_{\Omega^j},
\end{aligned}
\tag{D.6}
$$

where $\nabla_i = \partial_{\Omega^i} - \Omega_i\Omega^j\partial_{\Omega^j}$ is the covariant operator[48] on the unit sphere $S^{d-2}$, and the other

---

[48] In terms of the angular coordinates $\theta_1,\dots,\theta_{d-2}$ the covariant operator on the unit sphere is given by

$$
\nabla_1 = -\sin\theta_1\partial_{\theta_1}, \quad \nabla_2 = \cos\theta_1\cos\theta_2\partial_{\theta_1} - \frac{\sin\theta_2}{\sin\theta_1}\partial_{\theta_2}, \quad \dots,
$$

$$
\nabla_{d-2} = \sum_{j=1}^{d-3}\frac{\cos\theta_j\sin\theta_j\cdots\sin\theta_{d-3}\cos\theta_{d-2}}{\sin\theta_1\cdots\sin\theta_j}\partial_{\theta_j} - \frac{\sin\theta_{d-2}}{\sin\theta_1\cdots\sin\theta_{d-3}}\partial_{\theta_{d-2}},
$$

$$
\nabla_{d-1} = \sum_{j=1}^{d-3}\frac{\cos\theta_j\sin\theta_j\cdots\sin\theta_{d-3}\sin\theta_{d-2}}{\sin\theta_1\cdots\sin\theta_j}\partial_{\theta_j} + \frac{\cos\theta_{d-2}}{\sin\theta_1\cdots\sin\theta_{d-3}}\partial_{\theta_{d-2}}.
$$

$d + 1$ conformal generators take the form

$$iJ_{-10} = \frac{L^2 \cosh(t/L)}{\sqrt{L^2 - r^2}} \partial_t + \frac{r}{L} \sqrt{L^2 - r^2} \sinh(t/L) \partial_r$$

$$iJ_{-1i} = \frac{L^2 - r^2}{L} \Omega_i \partial_r + \frac{L}{r} \nabla_i \tag{D.7}$$

$$iJ_{-1d} = -\frac{L^2 \sinh(t/L)}{\sqrt{L^2 - r^2}} \partial_t - \frac{r}{L} \sqrt{L^2 - r^2} \cosh(t/L) \partial_r .$$

Finally, by comparing expression (A.6) for the conformal Killing vector that preserves a causal diamond in dS with the previous conformal generators, we see that $\zeta$ can be written as a linear combination of $J_{0d}$ and $J_{-10}$

$$\zeta^{\mathrm{dS}} = \frac{iL}{R} \left( J_{0d} - \sqrt{1 - (R/L)^2} J_{-10} \right) . \tag{D.8}$$

## D.2   Conformal Killing vectors of Anti-de Sitter space

Global coordinates for $d$-dimensional Anti-de Sitter space are defined by

$$X^{-1} = \sqrt{L^2 + r^2} \cos(t/L), \qquad X^d = L,$$
$$X^0 = \sqrt{L^2 + r^2} \sin(t/L), \qquad X^i = r \, \Omega^i, \quad i = 1, \ldots, d-1. \tag{D.9}$$

Note that here we have set a space coordinate $X^d$, instead of a time coordinate $X^{-1}$, equal to the curvature scale $L$. In this way the embedding constraint (D.1) turns into the familiar embedding equation for AdS, which is that of a hyperboloid in $\mathbb{R}^{2,d-1}$. It is manifest from this embedding that the isometry group of Lorentzian $\mathrm{AdS}_d$ is $O(2, d-1)$, whereas the full conformal group is the same as that of $\mathrm{dS}_d$, i.e. $O(2, d)$.

In terms of these coordinates, the induced metric on the light cone reads

$$ds^2 = -[1 + (r/L)^2] dt^2 + [1 + (r/L)^2]^{-1} dr^2 + r^2 d\Omega_{d-2}^2 . \tag{D.10}$$

For $r \geq 0$ and $0 \leq t < 2\pi L$ this solution covers the entire hyperboloid. However, to avoid closed timelike curves AdS is usually defined as the universal cover of the hyperboloid. This means that the timelike cycle is unwrapped, i.e. the range of the coordinate $t$ is extended: $-\infty < t < \infty$.

The generators of the conformal group take the following form in global AdS coordinates

$$iJ_{-10} = L \partial_t$$

$$iJ_{-1i} = -\frac{L r \Omega_i \sin(t/L)}{\sqrt{L^2 + r^2}} \partial_t + \sqrt{L^2 + r^2} \Omega_i \cos(t/L) \partial_r + \frac{\sqrt{L^2 + r^2} \cos(t/L)}{r} \nabla_i$$

$$iJ_{0i} = \frac{L r \Omega_i \cos(t/L)}{\sqrt{L^2 + r^2}} \partial_t + \sqrt{L^2 + r^2} \Omega_i \sin(t/L) \partial_r + \frac{\sqrt{L^2 + r^2} \sin(t/L)}{r} \nabla_i$$

$$iJ_{ij} = \Omega_j \partial_{\Omega^i} - \Omega_i \partial_{\Omega^j} \tag{D.11}$$

$$iJ_{-1d} = -\frac{L^2 \sin(t/L)}{\sqrt{L^2 + r^2}} \partial_t - \frac{r}{L} \sqrt{L^2 + r^2} \cos(t/L) \partial_r$$

$$iJ_{0d} = \frac{L^2 \cos(t/L)}{\sqrt{L^2 + r^2}} \partial_t - \frac{r}{L} \sqrt{L^2 + r^2} \sin(t/L) \partial_r$$

$$iJ_{id} = \frac{L^2 + r^2}{L} \Omega_i \partial_r + \frac{L}{r} \nabla_i .$$

The first four generators are true Killing vector fields, whereas the latter three are conformal Killing fields. The conformal Killing vector that preserves a causal diamond centered at the origin $r = 0$ of AdS can be obtained from (A.6) by analytically continuing the curvature scale $L \to iL$. One then finds the following expression in terms of the conformal generators

$$\zeta^{\text{AdS}} = \frac{iL}{R} \left( \sqrt{1 + (R/L)^2} J_{0d} - J_{-10} \right). \tag{D.12}$$

Notice that for a causal diamond of infinite size, i.e. $R/L \to \infty$, the conformal Killing field $\zeta$ reduces to the conformal generator $iJ_{0d}$. The $t = 0$ time slice of this diamond covers the entire AdS time slice, and at the center of this diamond $\zeta$ is equal to the timelike Killing vector $L\partial_t$.

Another useful set of coordinates for AdS are the Poincaré coordinates, which are defined by

$$X^{-1} = \frac{1}{2z} \left( L^2 - t^2 + \vec{x}^2 + z^2 \right), \qquad X^1 = \frac{1}{2z} \left( L^2 + t^2 - \vec{x}^2 - z^2 \right),$$
$$X^0 = Lt/z, \qquad X^i = Lx^i/z, \qquad X^d = L, \qquad i = 2, \ldots, d-1. \tag{D.13}$$

The induced metric in Poincaré coordinates is

$$ds^2 = \frac{L^2}{z^2} \left( -dt^2 + d\vec{x}^2 + dz^2 \right), \tag{D.14}$$

where the coordinates $t$ and $x^i$ range from $-\infty$ to $\infty$. The coordinate $z$ behaves as a radial coordinate and divides the hyperboloid into two charts ($z \lessgtr 0$). The Poincaré patch of AdS is typically taken to be the chart $z > 0$, which covers one half of the hyperboloid. The location $z = 0$ corresponds to the conformal boundary of AdS.

It is convenient to introduce the following generators of the conformal group

$$D = J_{-11}, \qquad P_\mu = J_{\mu-1} - J_{\mu 1},$$
$$M_{\mu\nu} = J_{\mu\nu}, \qquad K_\mu = J_{\mu-1} + J_{\mu 1}, \tag{D.15}$$

where $\mu = 0, i$, or $d$, corresponding to the coordinates $(t, x^i, z)$ respectively. On the conformal boundary of AdS these generators turn into the standard conformal generators of flat space (where $D$ denotes the generator of dilatations, $P_\mu$ of translations, $M_{\mu\nu}$ of Lorentz transformations, and $K_\mu$ of special conformal transformations). In Poincaré coordinates the conformal generators are equal to

$$iD = -\left( t\partial_t + x^i\partial_i + z\partial_z \right),$$
$$iP_t = -L\partial_t, \qquad iP_i = -L\partial_i, \qquad iP_z = -L\partial_z,$$
$$iK_t = \frac{-1}{L} \left[ (t^2 + \vec{x}^2 + z^2)\partial_t + 2t(x^i\partial_i + z\partial_z) \right],$$
$$iK_i = \frac{1}{L} \left[ (t^2 - \vec{x}^2 - z^2)\partial_i + 2x_i(t\partial_t + x^j\partial_j + z\partial_z) \right], \tag{D.16}$$
$$iK_z = \frac{1}{L} \left[ (t^2 - \vec{x}^2 + z^2)\partial_z + 2z(t\partial_t + x^i\partial_i) \right],$$
$$iM_{ti} = x_i\partial_t + t\partial_i, \qquad iM_{iz} = z\partial_x - x\partial_z,$$
$$iM_{tz} = z\partial_t + t\partial_z, \qquad iM_{ij} = x_j\partial_i - x_i\partial_j.$$

The generators $P_z, K_z, M_{tz}$ and $M_{iz}$ are conformal Killing vectors, and the other generators are true Killing vectors. With this list of conformal generators at hand, we are able to write the conformal Killing field (D.12) that preserves a diamond in AdS in terms of Poincaré coordinates

$$\zeta^{\text{AdS}} = \frac{iL}{R} \left[ \sqrt{1 + (R/L)^2} J_{0d} + J_{0-1} \right] = \frac{iL}{R} \left[ \sqrt{1 + (R/L)^2} M_{tz} + \frac{1}{2}(P_t + K_t) \right]$$
$$= \frac{1}{2R} \left[ \sqrt{L^2 + R^2}(2z\partial_t + 2t\partial_z) - (L^2 + t^2 + \vec{x}^2 + z^2)\partial_t - 2tx^i\partial_i - 2tz\partial_z \right]. \tag{D.17}$$

This conformal Killing vector field generates a modular flow inside a causal diamond $\mathcal{D}$ in the Poincaré patch of AdS. The center of the diamond $\mathcal{D}$ is located at $\{t = 0, z = L, x^i = 0\}$, and its boundaries are given by

$$\text{past and future vertices:} \quad p, p' = \left\{ t = \pm R, z = \sqrt{L^2 + R^2}, x^i = 0 \right\},$$

$$\text{edge/bifurcation surface:} \quad \mathcal{B} = \left\{ t = 0, z = \sqrt{L^2 + R^2} \pm \sqrt{R^2 - \vec{x}^2} \right\},$$

$$\text{past null boundary:} \quad \mathcal{H}_{\text{past}} = \left\{ z = \sqrt{L^2 + R^2} \pm \sqrt{(R + t)^2 - \vec{x}^2} \right\},$$

$$\text{future null boundary:} \quad \mathcal{H}_{\text{future}} = \left\{ z = \sqrt{L^2 + R^2} \pm \sqrt{(R - t)^2 - \vec{x}^2} \right\}.$$

One can readily check that the conformal Killing vector (D.17) vanishes at $p, p'$ and $\mathcal{B}$, and acts as a null flow on $\mathcal{H}_{\text{past/future}}$. Hence this causal diamond is entirely contained within the $z > 0$ Poincaré patch.

One can, of course, shift the causal diamond such that its center is located at a different position. An interesting special case is the diamond which is centered at the boundary of AdS, i.e. $z = 0$, and whose past and future null boundaries coincide with the Killing horizon of AdS-Rindler space. This can be established by sending $z \to z + \sqrt{L^2 + R^2}$ in (D.17). The conformal Killing vector then turns into the boost Killing vector of AdS-Rindler space, given in [17]

$$\xi^{\text{AdS-Rindler}} = \frac{iL}{2R} \left[ \left(1 - (R/L)^2\right) J_{0-1} + \left(1 + (R/L)^2\right) J_{01} \right] = \frac{iL}{2R} \left[ K_t - (R/L)^2 P_t \right]$$

$$= \frac{1}{2R} \left[ \left(R^2 - t^2 - \vec{x}^2 - z^2\right) \partial_t - 2t x^i \partial_i - 2t z \partial_z \right]. \tag{D.18}$$

Note that the normalization of the boost Killing vector is different than in [17], since we have set the surface gravity of the Killing horizon equal to one. The center of this diamond is located at $\{t = 0, z = 0, x^i = 0\}$, and the past and future vertices are at $\{t = \pm R, z = 0, x^i = 0\}$. Moreover, the bifurcation surface corresponds to the hemisphere $\mathcal{B} = \left\{ t = 0, z^2 + \vec{x}^2 = R^2 \right\}$. There exist two solutions to this quadratic equation, i.e. $z = \pm\sqrt{R^2 - \vec{x}^2}$, so there are in fact two equal-sized Rindler wedges in AdS that are preserved by this boost Killing vector (see Fig. 6a).

# E  Conformal transformation from Rindler space to causal diamond

It is well known in the literature [12, 22, 23] that there exists a special conformal transformation that maps Rindler space to a causal diamond in flat space. In this appendix we describe this transformation and derive the conformal Killing vector that preserves the diamond from the boost Killing vector of Rindler space. In particular, we focus on the conformal transformation from the right Rindler wedge to the diamond whose left edge is fixed at the bifurcation surface of the Rindler wedge. (See Fig. 4b for a depiction of this diamond inside the right Rindler wedge.) The special conformal transformation and its inverse are given by

$$x^\mu = \frac{X^\mu - X \cdot X C^\mu}{1 - 2X \cdot C + X \cdot X C \cdot C}, \qquad X^\mu = \frac{x^\mu + x \cdot x C^\mu}{1 + 2x \cdot C + x \cdot x C \cdot C}, \tag{E.1}$$

where $C^\mu = (0, -1/(2R), 0, 0)$, and $x^\mu$ and $X^\mu$ are Cartesian coordinates in Minkowski space. This is just a shift of the map mentioned in [12] (and we included a minus sign in $C^\mu$, correcting a typo in their paper). The transformation $X^\mu \to x^\mu$ maps the Minkowski metric to a metric on the diamond given by $ds^2 = \Omega^2 \eta_{\mu\nu} dx^\mu dx^\nu = \eta_{\mu\nu} dX^\mu dX^\nu$, where the conformal factor is given by

$$\Omega = 1 - 2X \cdot C + X \cdot X C \cdot C = [1 + 2x \cdot C + x \cdot x C \cdot C]^{-1}. \tag{E.2}$$

Note that the conformal transformation maps spatial infinity $i^0 = (T = 0, X^1 = \infty)$ to the right edge of the diamond $(t = 0, x^1 = 2R)$, and the origins are also mapped to each other.

Further, with the help of Mathematica, we find that under the mapping (E.1) the boost Killing field

$$\xi = X^1 \partial_T + T \partial_{X^1} \tag{E.3}$$

becomes the conformal Killing field that preserves the diamond

$$\zeta = \frac{1}{2R} \left[ \left( R^2 - (x^1 - R)^2 - (x^i)^2 - t^2 \right) \partial_t - 2t(x^1 - R)\partial_{x^1} - 2tx^i\partial_i \right], \tag{E.4}$$

where $i = 2, \ldots, d - 1$. By replacing $x^1$ by $x^1 + R$ one obtains the standard conformal Killing vector (A.7) that preserves a diamond centered at the origin of flat space. At $T = t = 0$ the norm of the two vector fields (E.3) and (E.4) is

$$|\xi| \overset{(T=0)}{=} X^1, \qquad |\zeta| \overset{(t=0)}{=} \frac{X^1}{\Omega} = x^1 - \frac{\vec{x}^2}{2R}. \tag{E.5}$$

The former blows up at spatial infinity $i^0$, whereas the latter vanishes at the edge of the diamond $(x^1 - R)^2 + (x^i)^2 = R^2$. This is because, although both $X^1$ and $\Omega$ diverge at spatial infinity, their ratio vanishes in this limit.

# F  Acceleration of the conformal Killing flow

In this appendix we show that the orbits of the conformal Killing field that preserves a maximally symmetric causal diamond have uniform acceleration inside the diamond. In the $(s, x)$ coordinate system, defined in Appendix B, the conformal Killing vector that preserves the diamond is $\zeta = \partial_s$. The flow lines of $\zeta$ coincide at the past and future tip of the diamond, but never cross inside the diamond (see Fig. 2). The velocity vector of the conformal Killing flow is $u^a = \zeta^a / \sqrt{-\zeta \cdot \zeta} = \delta_s{}^a C^{-1}$, and the acceleration vector is defined as $a^b = u^a \nabla_a u^b = \delta_x{}^b C^{-3} \partial_x C$. Using the expression (B.5) for $C = C(s, x)$, we find that the proper acceleration $a := \sqrt{a^b a_b}$ is equal to

$$a(x) = C^{-2} \partial_x C = R^{-1} \sinh x. \tag{F.1}$$

Note that the proper acceleration depends only on $x$, which is constant on the conformal Killing orbits.[49] The central orbit at $x = 0$ is unaccelerated, and at the edge where $x \to \infty$ the acceleration diverges.

Finally, we note that the surface gravity associated to $\zeta$ is equal to the redshifted proper acceleration evaluated at the bifurcation surface $\mathcal{B}$, i.e.

$$\kappa = \lim_{x \to \infty} Ca = 1. \tag{F.2}$$

This formula for the surface gravity is known to be equivalent to other definitions of surface gravity for Killing horizons (see e.g. [88] p. 332). For bifurcate conformal Killing horizons, following the same steps as [88], the acceleration definition (F.2) can be shown to be equivalent to the definition (C.1), but only in the limit as the bifurcation surface is approached, since the conformal Killing field is instantaneously a true Killing field at $\mathcal{B}$ (see Appendix C).

---

[49]We were unable to anticipate this surprising fact.

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
