# Peer review of "Gravitational Thermodynamics of Causal Diamonds in (A)dS"

_SciPost Physics, doi:SciPost Phys. 7, 079 (2019)_

## Round 2 · Referee Report · Anonymous (Referee 1) · 2019-10-18

Strengths

1) interesting, with many potential ramifications 2) thorough and complete 3) clearly written, except in a few odd spots 4) candid - when the authors don't know something, they say it

Weaknesses

1) nothing, except possibly that it's lengthy -1) but this length was probably unavoidable

Report

This paper tells everything anyone will need to know about the thermodynamics of causal diamonds in symmetric spaces. There is nothing I could possibly suggest to improve about this manuscript because it is interesting, comprehensive, clear and honest. I suggest publication essentially without changes.

Requested changes

The only place that I found confusing was the two paragraphs below equation (4.8). The explanation why matter can appear both as an energy or an entropy variation, and which terms are explicit and/or implicit, is rather disorienting.

I would suggest clarifying this fragment. This is a request and not a demand; it's entirely optional.

  • validity: top
  • significance: top
  • originality: high
  • clarity: high
  • formatting: perfect
  • grammar: perfect

Author:  Theodore Jacobson  on 2019-12-09  [id 669]

(in reply to Report 1 on 2019-10-18)

We thank the referee for suggesting we clarify why matter can appear explicitly in the first law either via its energy variation or via its entropy variation, but not both. This is indeed an important point. We have revised the presentation, attempting to explain it more clearly.

---

## Editorial Decision

published